# ACCURATE LINK PREDICTION VIA PU-LEARNING

## ABSTRACT

Given an edge-incomplete graph, how can we accurately find the missing links? The link prediction in edge-incomplete graphs aims to discover the missing relations between entities when their relationships are represented as a graph. Edge-incomplete graphs are prevalent in real-world due to practical limitations, such as not checking all users when adding friends in a social network. Addressing the problem is crucial for various tasks, including recommending friends in social networks and finding references in citation networks. However, previous approaches for link prediction rely heavily on the given edge-incomplete (observed) graph, making it challenging to consider the missing (unobserved) links during training. In this paper, we propose PULL (PU-LEARNING-BASED LINK PREDICTOR), an accurate link prediction method based on the positive-unlabeled (PU) learning. PULL treats the observed edges in the training graph as positive examples, and the unconnected node pairs as unlabeled ones. PULL effectively prevents the link predictor from overfitting to the observed graph by proposing latent variables for every edge, and leveraging the expected graph structure with respect to the variables. Extensive experiments on five real-world datasets show that PULL consistently outperforms the baselines for predicting links in edge-incomplete graphs.

## 1 INTRODUCTION

Given an edge-incomplete graph, how can we accurately find the missing links among the unconnected node pairs? Edge-incomplete graphs are easily encountered in real-world networks. In social networks, connections between users can be missing since we do not check every user when adding friends. In the context of citation networks, there may be missing citations as we do not review all published papers for citation. The objective of the link prediction in edge-incomplete graphs is to discover the undisclosed relationships between examples when we are provided with a graph that represents the known relationships among them (Liben-Nowell & Kleinberg, 2007). Such scenarios include finding uncited references in citation networks (Shibata et al., 2012; Liu et al., 2019), and recommending new friends in social networks (Wang et al., 2015; Daud et al., 2020a).

The main limitation of previous works (Kipf & Welling, 2016b; 2017; Pan et al., 2018; Zhang & Bai, 2023) for link prediction problem is that they rely strongly on the given edge-incomplete graph. They presume the edges of the given graph are fully observed ones, and do not consider the unobserved missing links while training. However, this does not always reflect the real-world scenarios where the presence of missing edges is frequently observed. This limits the model's ability to propagate information through unconnected node pairs, which may potentially form edges, overfitting a link predictor to the given edge-incomplete graph. Thus, it is important to consider the uncertainties of the given graph to obtain accurate linking probabilities between nodes.

In this work, we propose PULL (PU-LEARNING-BASED LINK PREDICTOR), an accurate method for link prediction in edge-incomplete graphs. To account for the uncertainties in the given graph structure while training a link predictor $f$, PULL exploits PU learning (see Section 2.2 for details). We treat the observed edges within the graph as positive examples and the unconnected node pairs, which may contain hidden connections, as unlabeled examples. We then construct an expected graph $\bar{\mathcal{G}}$ while proposing latent variables for the unlabeled (unconnected) node pairs to consider the hidden connections among them. This enables us to effectively propagate information through the unconnected edges, improving the prediction accuracy of $f$. Since the estimated linking probabilities of $f$ are prior knowledge for constructing the expected graph structure $\bar{\mathcal{G}}$, improved link predictor $f$ enhances the quality of $\bar{\mathcal{G}}$. Thus, PULL employs an iterative learning approach with two-steps to

achieve a repeated improvement of the link predictor $f$: a) constructing an expected graph structure $\bar{\mathcal{G}}$ based on the linking probabilities between nodes from the link predictor $f$, and b) training $f$ exploiting the expected graph $\bar{\mathcal{G}}$. Note that the updated $f$ is used to update $\bar{\mathcal{G}}$ in the next iteration.

Our contributions are summarized as follows:

- **Method.** We propose PULL, an accurate method for link prediction in graphs. PULL effectively overcomes a primary limitation of previous methods, which is their heavy reliance on the provided graph structure. This is achieved by training a link predictor with an expected graph structure while treating the unconnected edges as unlabeled ones.
- **Theory.** We theoretically analyze PULL, studying its relationship with the EM algorithm.
- **Experiments.** We perform various experiments on five real-world datasets, and show that PULL achieves the state-of-the-art link prediction performance.

The code and datasets are available at `https://github.com/graphmaster2023/pull`. The symbols used in this paper is in Table 2 (Appendix B.1).

## 2 RELATED WORKS AND PROBLEM DEFINITION

### 2.1 LINK PREDICTION IN GRAPHS

Link prediction in graphs has garnered significant attention in recent years, due to its successful application in various domains including social networks (Backstrom & Leskovec, 2011; Wang et al., 2016a; Daud et al., 2020b), recommendation systems (Afoudi et al., 2023; Kurt et al., 2019), and biological networks (Sulaimany et al., 2018; Long et al., 2022). Previous approaches for link prediction are categorized into two groups: embedding-based and autoencoder-based approaches.

Embedding-based approaches strive to create compact representations of nodes within a graph via random walk or propagation. These representations are subsequently employed to estimate the probability of connections between nodes. Deepwalk (Perozzi et al., 2014) and Node2Vec (Grover & Leskovec, 2016) create embeddings by simulating random walks on the graph. The concept is to generate embeddings in a way that nodes frequently appearing together in these random walks end up having similar representations. GCN (Kipf & Welling, 2017), LINE (Xu, 2017), GraphSAGE (Hamilton et al., 2017), and GAT (Velickovic et al., 2017) aggregate information from neighboring nodes to learn the embeddings, assuming adjacent nodes are similar. SEAL (Zhang & Chen, 2018) extends the link prediction problem into a subgraph classification problem.

The autoencoder-based methods exploit autoencoders to train a link predictor. GAE (Kipf & Welling, 2016a) is an autoencoder-based unsupervised framework for link prediction. VGAE is a variational graph autoencoder, which is a variant of GAE. VGAE explicitly models the uncertainty by introducing a probabilistic layer. ARGA and ARGVA (Pan et al., 2018) exploit adversarial training strategy to improve the performance of GAE and VGAE, respectively. GNAE and VGNAE (Ahn & Kim, 2021) found that autoencoder-based methods produce embeddings that converge to zero for isolated nodes, regardless of their input features. They utilize L2-normalization to get better embeddings for these isolated nodes. However, those embedding-based and autoencoder-based approaches assume that the edges of the given graph are fully observed. This overfits the node embeddings to the given edge-incomplete graph, degrading the link prediction performance.

### 2.2 PU LEARNING

The objective of PU (Positive-Unlabeled) learning is to train a binary classifier that effectively distinguishes positive and negative instances when only positive and unlabeled examples are available. Many algorithms are developed to address the uncertainty introduced by the lack of labeled negative examples. Unbiased risk estimator (URE) (du Plessis et al., 2014) considers the probability that each unlabeled example is a positive instance and adjusts the risk estimate accordingly. Non-negative risk estimator (Kiryo et al., 2017) improves the classification accuracy of URE by preventing the risk of unlabeled instances from going negative. However, those risk-based approaches require the ratio of real positive examples (class prior) among the whole ones in advance, which is not realistic.

Many graph-based PU learning approaches have been studied recently (Li et al., 2016; Zhang et al., 2019; Wu et al., 2019). PU-LP (Ma & Zhang, 2017) finds relatively positive examples from the

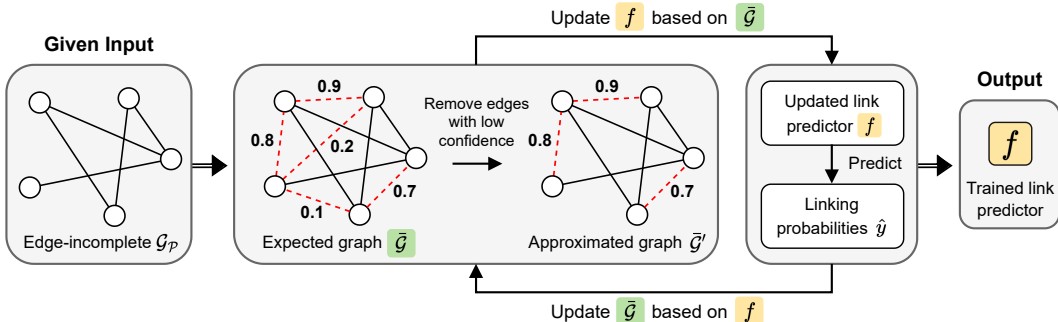

Figure 1: Overall structure of PULL. Given an edge-incomplete graph $\mathcal{G}_\mathcal{P}$ with a set $\mathcal{P}$ of observed edges, PULL first computes the expected graph structure $\bar{\mathcal{G}}$ by proposing latent variables for the edges. Then PULL utilizes $\bar{\mathcal{G}}$ to update the link predictor $f$. The marginal linking probabilities $\hat{y}$ obtained by the updated $f$ are used to compute $\bar{\mathcal{G}}$ in the next iteration.

unlabeled ones utilizing the given graph structure, and treats the rest as relatively negative ones. GRAB (Yoo et al., 2021) is the first approach to solve the graph-based PU learning problem without knowing the class prior in advance. However, those graph-based PU learning methods cannot be directly used in the link prediction problem since they aim to classify nodes, not edges, while considering the edges of the given graph as fully observed ones. Hao (2021) and Gan et al. (2022) proposed PU learning frameworks for link prediction considering the given edges as observed positive examples. However, their link prediction performance is constrained by the propagation of information through the edge-incomplete graph for obtaining node and edge representations.

## 3 PROPOSED METHOD

We propose PULL (PU-LEARNING-BASED LINK PREDICTOR), an accurate method for link prediction in edge-incomplete graphs. PULL effectively exploits the missing links for training the link predictor based on PU-learning approach. We illustrate the entire process of PULL in Figure 1 and Algorithm 1. The main challenges and our approaches are as follows:

1. **How can we consider the missing links while training?** We treat the given edges as observed positive examples, and the rest as unlabeled ones. We then propagate information through an expected graph structure by proposing latent variables to the unconnected node pairs. (Section 3.1).

2. **How can we effectively model the expected graph structure?** Computing the expected graph structure is computationally expensive since there are $2^{|\mathcal{E}_\mathcal{U}|}$ possible graph structures where $\mathcal{E}_\mathcal{U}$ is the set of unconnected edges. We effectively compute the expectation of graph structure by carefully designing the probabilities of graphs (Section 3.2).

3. **How can we gradually improve the performance of the link predictor?** PULL iteratively improves the quality of the expected graph structure, which is the evidence for training the link predictor (Section 3.3).

### 3.1 MODELING MISSING LINKS

In the problem of link prediction in edge-incomplete graph, we are given a feature matrix $\mathbf{X}$ and an edge-incomplete graph $\mathcal{G}_\mathcal{P}$ consisting of two sets of edges, $\mathcal{E}_\mathcal{P}$ and $\mathcal{E}_\mathcal{U}$. The set $\mathcal{E}_\mathcal{P}$ contains observed edges, and $\mathcal{E}_\mathcal{U}$ consists of unconnected node pairs; $\mathcal{E}_\mathcal{P} \cup \mathcal{E}_\mathcal{U}$ is a set of all possible node pairs. Then we aim to find unobserved connected edges among $\mathcal{E}_\mathcal{U}$ accurately (see Appendix A for a formal problem definition). Existing link prediction methods treat the given edges of $\mathcal{E}_\mathcal{P}$ as fully observed ones, and they propagate information through it to train a link predictor $f$. This overfits $f$ to the edge-incomplete graph, degrading the link prediction performance.

To prevent the overfitting problem of $f$ to the edge-incomplete graph, PULL models the given graph based on PU-learning approach. Since there are hidden connections in $\mathcal{E}_\mathcal{U}$, we treat the unconnected edges in $\mathcal{E}_\mathcal{U}$ as unlabeled examples, and the observed edges in $\mathcal{E}_\mathcal{P}$ as positive ones. Then we propose a latent variable $z_{ij} \in \{1, 0\}$ for every edge $e_{ij}$, indicating whether there is a link between nodes $i$

---

**Algorithm 1:** PULL (PU-LEARNING-BASED LINK PREDICTOR).

---

**Input** : Edge-incomplete graph $\mathcal{G}_\mathcal{P} = (\mathcal{V}, \mathcal{E}_\mathcal{P})$, feature matrix $\mathbf{X}$, set $\mathcal{E}_\mathcal{U}$ of unconnected edges, hyperparameter $r$, and link predictor $f_\theta(i,j)$ parameterized by $\theta$

**Output** : Best parameters $\theta$ of link predictor $f_\theta(i,j)$

---

1 Randomly initialize $\theta^{\text{new}}$ , and initialize $K$ as $|\mathcal{E}_\mathcal{P}|$;
2 **repeat**
3     $\theta \leftarrow \theta^{\text{new}}$;
4     $\bar{\mathcal{G}} \leftarrow \mathbb{E}_{\mathbf{z} \sim p(\mathbf{z} | \mathbf{X}, \mathcal{E}_\mathcal{P}, \theta)}[\mathbf{A}(\mathbf{z})] = \mathbf{A}^{\bar{\mathcal{G}}}$ ;         // compute expected graph structure by Equations (3), (4)
5     Approximate $\bar{\mathcal{G}}$ to $\bar{\mathcal{G}}'$ by selecting top-$K$ edges with the largest weights;
6     $K \leftarrow K + |\mathcal{E}_\mathcal{P}| * r$ ;
7     $\theta^{\text{new}} \leftarrow \arg\min_\theta \mathcal{L}(\theta; \bar{\mathcal{G}}', \mathbf{X})$;         // update $\theta$ using Equations (6), (7)
8 **until** *the maximum number of iterations is reached or the early stopping condition is met*;

---

and $j$ to consider the hidden connections; $z_{ij} = 1$ for every $e_{ij} \in \mathcal{E}_\mathcal{P}$, but not always $z_{ij} = 0$ for $e_{ij} \in \mathcal{E}_\mathcal{U}$. We denote the graph $\mathcal{G}_\mathcal{P}$ with latent variable $\mathbf{z} = \{z_{ij} \text{ for } e_{ij} \in (\mathcal{E}_\mathcal{P} \cup \mathcal{E}_\mathcal{U})\}$ as $\mathcal{G}_\mathcal{P}(\mathbf{z})$.

A main challenge is that we cannot propagate information through the *variablized graph* $\mathcal{G}_\mathcal{P}(\mathbf{z})$ while training $f$ since every edge $e_{ij} \in \mathcal{E}_\mathcal{U}$ of $\mathcal{G}_\mathcal{P}(\mathbf{z})$ is probabilistically connected. Instead, PULL exploits the expected graph structure $\bar{\mathcal{G}}$ over the latent variables $\mathbf{z}$. This enables us to train a link predictor $f$ accurately, considering the hidden connections in $\mathcal{E}_\mathcal{U}$. Since the link predictor gives prior knowledge for constructing the expected graph, improved $f$ enhances the quality of $\bar{\mathcal{G}}$. Thus, PULL trains the link predictor $f$ by iteratively performing the two steps: a) constructing an expected graph structure $\bar{\mathcal{G}}$ given marginal linking probabilities of trained $f$, and b) updating the link predictor $f$ using $\bar{\mathcal{G}}$, which is used to improve the quality of the expected graph in the next iteration.

## 3.2 EXPECTATION OF GRAPH STRUCTURE

PULL propagates information through the expected graph structure $\bar{\mathcal{G}}$ of $\mathcal{G}_\mathcal{P}(\mathbf{z})$ over the latent variable $p(\mathbf{z} \mid \mathbf{X}, \mathcal{E}_\mathcal{P}, \theta)$ while training a link predictor $f_\theta$ with learnable parameter $\theta$. $\bar{\mathcal{G}}$ requires computing the joint probabilities $p(\mathbf{z} \mid \mathbf{X}, \mathcal{E}_\mathcal{P}, \theta)$ for all possible graph structures $\mathcal{G}_\mathcal{P}(\mathbf{z})$. This is intractable since there are $2^{|\mathcal{E}_\mathcal{U}|}$ possible states of $\mathbf{z}$ in $\mathcal{G}_\mathcal{P}(\mathbf{z})$. Instead, PULL efficiently computes the expected graph structure by carefully designing the joint probability $p(\mathbf{z} \mid \mathbf{X}, \mathcal{E}_\mathcal{P}, \theta)$.

We convert the graph $\mathcal{G}_\mathcal{P}(\mathbf{z})$ with latent variables $\mathbf{z}$ into a line graph $L(\mathcal{G}_\mathcal{P}(\mathbf{z})) = (\mathcal{V}_L, \mathcal{E}_L)$ where nodes in $L(\mathcal{G}_\mathcal{P}(\mathbf{z}))$ represent the edges of $\mathcal{G}_\mathcal{P}(\mathbf{z})$, and two nodes in $L(\mathcal{G}_\mathcal{P}(\mathbf{z}))$ are connected if their corresponding edges in $\mathcal{G}_\mathcal{P}(\mathbf{z})$ are adjacent. Note that $\mathcal{V}_L$ contains both $\mathcal{E}_\mathcal{P}$ and $\mathcal{E}_\mathcal{U}$ of $\mathcal{G}_\mathcal{P}(\mathbf{z})$ since every node pair $(i,j)$ in $\mathcal{G}_\mathcal{P}(\mathbf{z})$ is correlated with variable $z_{ij}$. We then consider the line graph as a pairwise Markov network, which assumes that any two random variables in the network are conditionally independent of each other given the rest of the variables if they are not directly connected (Parsons, 2011). This simplifies the probabilistic modeling on graph-structured random variables, and effectively marginalizes the joint distribution of nodes in $L(\mathcal{G}_\mathcal{P}(\mathbf{z}))$, which corresponds to the distribution $p(\mathbf{z} \mid \mathbf{X}, \mathcal{E}_\mathcal{P}, \theta)$ of edges in $\mathcal{G}_\mathcal{P}(\mathbf{z})$.

With the Markov property, the joint distribution of nodes in the line graph $L(\mathcal{G}_\mathcal{P}(\mathbf{z})) = (\mathcal{V}_L, \mathcal{E}_L)$ is computed by the multiplication of all the node and edge potentials:

$$p(\mathbf{z} \mid \mathbf{X}, \mathcal{E}_\mathcal{P}, \theta) = \frac{1}{F} \prod_{ij \in \mathcal{V}_L} \phi_{ij}(z_{ij} \mid \mathbf{X}, \theta) \prod_{(ij,jk) \in \mathcal{E}_L} \psi_{ij,jk}(z_{ij}, z_{jk} \mid \mathbf{X}, \theta) \tag{1}$$

where $\phi_{ij}$ and $\psi_{ij,jk}$ are node and edge potentials for each transformed node $ij$ and edge $(ij, jk)$, respectively. The node potential $\phi_{ij}$ represents the unnormalized marginal linking probability between nodes $i$ and $j$ in the original graph $\mathcal{G}_\mathcal{P}(\mathbf{z})$. The edge potential $\psi_{ij,jk}$ denotes a degree of homophily between the edges containing a common node in $\mathcal{G}_\mathcal{P}(\mathbf{z})$. $F$ is the normalizing factor that ensures the distribution adds up to one. For simplicity, we omit $\mathbf{X}$ in $\phi_{ij}$ and $\psi_{ij,jk}$ in the rest of the paper.

We define the node potential $\phi_{ij}$ of $L(\mathcal{G}_\mathcal{P}(\mathbf{z}))$ as follows to make nodes in $\mathcal{G}_\mathcal{P}(\mathbf{z})$ with similar hidden representations have a higher likelihood of connection:

$$\phi_{ij}(z_{ij} = 1 \mid \theta) = \begin{cases} 1 & \text{if } e_{ij} \in \mathcal{E}_\mathcal{P} \\ f_\theta(i,j) = \text{sigmoid}(h_i \cdot h_j) & \text{otherwise} \end{cases}$$

where $h_i$ is the hidden representation of node $i$ in $\mathcal{G}_{\mathcal{P}}(\mathbf{z})$ parameterized by $\theta$, and $\phi_{ij}(z_{ij} = 0 \mid \theta) = 1 - \phi_{ij}(z_{ij} = 1 \mid \theta)$. We use a GCN that propagates information through $\bar{\mathcal{G}}$ followed by a sigmoid function as $f_\theta(i, j)$. We set $\phi_{ij}(z_{ij} = 1 \mid \theta) = 1$ for $e_{ij} \in \mathcal{E}_{\mathcal{P}}$ since the linking probability of an observed edge of $\mathcal{G}_{\mathcal{P}}$ is 1. We define $\psi_{ij,jk}$ as a constant $c$ to make the joint distribution focus on the marginal linking probabilities. Then the normalizing constant $F$ in Equation (1) becomes $c^{|\mathcal{E}_L|}$ since $\sum_{\mathbf{z}} \prod_{ij \in \mathcal{V}_L} \phi_{ij}(z_{ij} \mid \theta) = 1$ (see Lemma 1 in Appendix C for proof). As a result, the joint probability $p(\mathbf{z} \mid \mathbf{X}, \mathcal{E}_{\mathcal{P}}, \theta)$ is expressed by the multiplication of node potentials of the line graph:

$$p(\mathbf{z} \mid \mathbf{X}, \mathcal{E}_{\mathcal{P}}, \theta) = \prod_{ij \in \mathcal{V}_L} \phi_{ij}(z_{ij} \mid \theta) = \prod_{e_{ij} \in (\mathcal{E}_{\mathcal{P}} \cup \mathcal{E}_{\mathcal{U}})} \phi_{ij}(z_{ij} \mid \theta) = \prod_{e_{ij} \in \mathcal{E}_{\mathcal{U}}} \phi_{ij}(z_{ij} \mid \theta). \quad (2)$$

Using the marginalized joint probability $p(\mathbf{z} \mid \mathbf{X}, \mathcal{E}_{\mathcal{P}}, \theta)$ in Equation (2), we express the expected graph structure $\bar{\mathcal{G}}$ with regard to the latent variables $\mathbf{z}$. Let $\mathbf{A}(\mathbf{z})$ be the adjacency matrix representing the state $\mathbf{z}$ where the $(i, j)$-th component of $\mathbf{A}(\mathbf{z})$, which we denote as $\mathbf{A}(\mathbf{z})_{ij}$, is $z_{ij} \in \{1, 0\}$. Then the corresponding weighted adjacency matrix $\mathbf{A}^{\bar{\mathcal{G}}}$ of the expected graph $\bar{\mathcal{G}}$ is computed as follows:

$$\mathbf{A}^{\bar{\mathcal{G}}} = \mathbb{E}_{\mathbf{z} \sim p(\mathbf{z}|\mathbf{X}, \mathcal{E}_{\mathcal{P}}, \theta)}[\mathbf{A}(\mathbf{z})] = \sum_{\mathbf{z}} p(\mathbf{z} \mid \mathbf{X}, \mathcal{E}_{\mathcal{P}}, \theta) \mathbf{A}(\mathbf{z}) = \sum_{\mathbf{z}} \prod_{e_{ij} \in \mathcal{E}_{\mathcal{U}}} \phi_{ij}(z_{ij} \mid \theta) \mathbf{A}(\mathbf{z}). \quad (3)$$

The $(i, j)$-th component $\mathbf{A}^{\bar{\mathcal{G}}}_{ij}$ of $\mathbf{A}^{\bar{\mathcal{G}}}$ is simply expressed as follows:

$$\mathbf{A}^{\bar{\mathcal{G}}}_{ij} = \phi_{ij}(z_{ij} = 1 \mid \theta) \sum_{\mathbf{z}|z_{ij}=1} \prod_{e_{kl} \in \mathcal{E}_{\mathcal{U}} \setminus \{e_{ij}\}} \phi_{kl}(z_{kl} \mid \theta) \mathbf{A}(\mathbf{z})_{ij} = \phi_{ij}(z_{ij} = 1 \mid \theta) \quad (4)$$

since $\mathbf{A}(\mathbf{z})_{ij} = 1$ for $z_{ij} = 1$, and $\sum_{\mathbf{z}|z_{ij}=1} \prod_{e_{kl} \in \mathcal{E}_{\mathcal{U}} \setminus \{e_{ij}\}} \phi_{kl}(z_{kl} \mid \theta) = 1$ (see Lemma 1 in Appendix C for proof). As a result, we simply express the expected graph $\bar{\mathcal{G}}$ by an weighted adjacency matrix $\mathbf{A}^{\bar{\mathcal{G}}}$ where $\mathbf{A}^{\bar{\mathcal{G}}}_{ij} = \phi_{ij}(z_{ij} = 1 \mid \theta)$.

Using the expected graph $\bar{\mathcal{G}}$ directly to train the link predictor $f$ may lead to oversmoothing problem, as $\bar{\mathcal{G}}$ is a fully connected graph represented by $\mathbf{A}^{\bar{\mathcal{G}}}$. Moreover, the training time increases exponentially with the number of nodes. To address these challenges, PULL utilizes an approximated one of $\bar{\mathcal{G}}$ for training $f$, which contains edges with high confidence. Specifically, we approximate $\bar{\mathcal{G}}$ by keeping the top-$K$ edges with the largest weights, while removing the rest. We refer to this approximated one as $\bar{\mathcal{G}}'$, and its corresponding adjacency matrix as $\mathbf{A}^{\bar{\mathcal{G}}'}$. From the perspective of PU learning, selecting edges in $\bar{\mathcal{G}}$ can be viewed as selecting relatively connected edges among the unlabeled ones, while treating the rest as relatively unconnected edges. We gradually increase the number $K$ of selected edges in proportion to that of observed edges through the iterations, which is expressed by $K \leftarrow K + r|\mathcal{E}_{\mathcal{P}}|$, giving more trust in the expected graph structure $\bar{\mathcal{G}}$. This is because the quality of $\bar{\mathcal{G}}$ improves through the iterations (see Figure 2). We set $r = 0.05$ in our experiments.

## 3.3 Iterative Learning

At each iteration, PULL computes the expected graph $\bar{\mathcal{G}}$ with respect to $p(\mathbf{z} \mid \mathbf{X}, \mathcal{E}_{\mathcal{P}}, \theta)$ given a trained link predictor $f_\theta$ with current parameter $\theta$. Then PULL propagates information through $\bar{\mathcal{G}}'$ instead of the given edge-incomplete graph $\mathcal{G}_{\mathcal{P}}$ to train a new link predictor $f_\theta$ with new parameter $\theta^{\text{new}}$. This prevents PULL from overfitting to $\mathcal{G}_{\mathcal{P}}$, thus improving the link prediction performance.

To optimize the new parameter $\theta^{\text{new}}$, we propose the binary cross entropy loss $\mathcal{L}_E$ in Equation (5) by treating the given edges in $\mathcal{E}_{\mathcal{P}}$ and the unconnected edges in $\mathcal{E}_{\mathcal{U}}$ as positive and unlabeled (PU) examples, respectively. For the unconnected edges, we use the expected linking probability $\mathbf{A}^{\bar{\mathcal{G}}'}_{ij}$, which are obtained from the current link predictor $f_\theta$, as pseudo labels for each $e_{ij}$:

$$
\begin{aligned}
\mathcal{L}_E &= - \sum_{e_{ij} \in \mathcal{E}_{\mathcal{P}}} \log \hat{y}_{ij} - \sum_{e_{ij} \in \mathcal{E}_{\mathcal{U}}} \left( \mathbf{A}^{\bar{\mathcal{G}}'}_{ij} \log \hat{y}_{ij} + (1 - \mathbf{A}^{\bar{\mathcal{G}}'}_{ij}) \log(1 - \hat{y}_{ij}) \right) \\
&= - \sum_{e_{ij} \in \mathcal{E}_{\mathcal{P}}} \log \hat{y}_{ij} - \sum_{e_{ij} \in \mathcal{E}^r_{\mathcal{P}}} \left( \mathbf{A}^{\bar{\mathcal{G}}'}_{ij} \log \hat{y}_{ij} + (1 - \mathbf{A}^{\bar{\mathcal{G}}'}_{ij}) \log(1 - \hat{y}_{ij}) \right) - \sum_{e_{ij} \in \mathcal{E}^r_{\mathcal{U}}} \log(1 - \hat{y}_{ij})
\end{aligned} \quad (5)
$$

where $\hat{y}_{ij} = f_{\theta^{\text{new}}}(i, j)$. $\mathcal{E}^r_{\mathcal{P}}$ is the set of relatively connected edges selected from $\mathcal{E}_{\mathcal{U}}$ when approximating the expected graph structure $\bar{\mathcal{G}}$ to $\bar{\mathcal{G}}'$ in Section 3.2, and $\mathcal{E}^r_{\mathcal{U}} = \mathcal{E}_{\mathcal{U}} \setminus \mathcal{E}^r_{\mathcal{P}}$.

However, in real-world graphs, there is a severe imbalance between the numbers of connected edges and unconnected ones. We balance them by randomly sampling $|\mathcal{E}_{\mathcal{P}} \cup \mathcal{E}^r_{\mathcal{P}}|$ unconnected edges among

$\mathcal{E}_{\mathcal{U}}^r$ for every epoch. Then Equation (5) is written as follows:

$$\mathcal{L}'_E = - \sum_{e_{ij} \in \mathcal{E}_{\mathcal{P}}} \log \hat{y}_{ij} - \sum_{e_{ij} \in \mathcal{E}_{\mathcal{P}}^r} \left( \mathbf{A}_{ij}^{\bar{\mathcal{G}}'} \log \hat{y}_{ij} + (1 - \mathbf{A}_{ij}^{\bar{\mathcal{G}}'}) \log(1 - \hat{y}_{ij}) \right) - \sum_{e_{ij} \in \mathcal{E}_{\mathcal{U}}^s} \log(1 - \hat{y}_{ij}) \quad (6)$$

where $\mathcal{E}_{\mathcal{U}}^s$ is the set of randomly sampled edges among $\mathcal{E}_{\mathcal{U}}^r$ with size $|\mathcal{E}_{\mathcal{U}}^s| = |\mathcal{E}_{\mathcal{P}} \cup \mathcal{E}_{\mathcal{P}}^r|$.

If the current parameter $\theta$ of the link predictor are inaccurate, the quality of the expected graph structure deteriorates, leading to the next iteration's parameter $\theta^{\text{new}}$ becoming even more inaccurate. Thus, we propose another loss term $\mathcal{L}_C$ for correction, which measures the binary cross entropy for all observed edges and randomly sampled unconnected edges from $\mathcal{E}_{\mathcal{U}}$:

$$\mathcal{L}_C = - \sum_{e_{ij} \in \mathcal{E}_{\mathcal{P}}} \log \hat{y}_{ij} - \sum_{e_{kl} \in \mathcal{E}'_{\mathcal{U}}} \log(1 - \hat{y}_{ij}). \quad (7)$$

where $\mathcal{E}'_{\mathcal{U}}$ is the set of randomly sampled node pairs from $\mathcal{E}_{\mathcal{U}}$ with size $|\mathcal{E}'_{\mathcal{U}}| = |\mathcal{E}_{\mathcal{P}}|$. $\mathcal{L}_C$ effectively prevents excessive self-reinforcement in the link predictor of PULL (see Figure 4).

As a result, PULL finds the best parameter $\theta^{\text{new}}$ for each iteration by minimizing the sum of the two loss terms in Equations (6), (7). We denote the final loss function as $\mathcal{L}(\theta^{\text{new}}; \bar{\mathcal{G}}', \mathbf{X}) = \mathcal{L}'_E + \mathcal{L}_C$. The new parameter $\theta^{\text{new}}$ is used as the current parameter $\theta$ for the next iteration. The iterations stop if the maximum number of iterations is reached or the early stopping condition (see Section 4.1) is met.

### 3.4 Theoretical Analysis

We theoretically analyze the connection between PULL and the EM (Expectation-Maximization) algorithm. EM is an iterative method used for estimating model parameter $\theta$ when there are missing or unobserved data. It assigns latent variables $\mathbf{z}$ to the unobserved data, and maximizes the expectation of the log likelihood $\log p(\mathbf{y}, \mathbf{z} \mid \mathbf{X}, \theta)$ in terms of $\mathbf{z}$ to optimize $\theta$ where $\mathbf{y}$ and $\mathbf{X}$ are target and input variables, respectively.

In our problem, the target variables are represented as $\mathcal{E}_{\mathcal{P}}$. Thus, the expectation of the log likelihood given current parameter $\theta$ is written as follows:

$$Q(\theta^{\text{new}} \mid \theta) = \mathbb{E}_{\mathbf{z} \sim p(\mathbf{z}|\mathbf{X}, \mathcal{E}_{\mathcal{P}}, \theta)}[\log p(\mathcal{E}_{\mathcal{P}}, \mathbf{z} \mid \mathbf{X}, \theta^{\text{new}})] \quad (8)$$

where $\theta^{\text{new}}$ is the new parameter. The EM algorithm finds $\theta^{\text{new}}$ that maximizes $Q(\theta^{\text{new}} \mid \theta)$, and they are used as $\theta$ in the next iteration. The algorithm is widely used in situations involving latent variables since it always improves the likelihood $Q(\theta^{\text{new}} \mid \theta)$ through the iterations (Murphy, 2012).

PULL iteratively optimizes the parameter $\theta$ of a link predictor $f$ by minimizing both $\mathcal{L}'_E$ and $\mathcal{L}_C$ where $\mathcal{L}'_E$ is the approximation of $\mathcal{L}_E$ in Equation (5). We compare Equations (5) and (8) to show the similarity between the iterative minimization of $\mathcal{L}_E$ in PULL and the iterative maximization of $Q(\theta^{\text{new}} \mid \theta)$ in the EM algorithm.

PULL effectively expresses the distribution $p(\mathbf{z} \mid \mathbf{X}, \mathcal{E}_{\mathcal{P}}, \theta)$ in Equation (8) by the multiplication of node potentials in Equation (2). For the joint probability $p(\mathcal{E}_{\mathcal{P}}, \mathbf{z} \mid \mathbf{X}, \theta^{\text{new}})$ in Equation (8), we approximate it using a link predictor $f_{\theta^{\text{new}}}$ with new parameter $\theta^{\text{new}}$. We consider the link predictor $f_{\theta_{\text{new}}}$ as a marginalization function that gives marginal linking probabilities for each node pair. We also assume that the marginal distributions obtained by $f_{\theta_{\text{new}}}$ are mutually independent. Then the joint probability $p(\mathcal{E}_{\mathcal{P}}, \mathbf{z} \mid \mathbf{X}, \theta^{\text{new}})$ is approximated as follows:

$$p(\mathcal{E}_{\mathcal{P}}, \mathbf{z} \mid \mathbf{X}, \theta^{\text{new}}) \approx \prod_{e_{ij} \in \mathcal{E}_{\mathcal{P}}} \hat{y}_{ij} \prod_{e_{ij} \in \mathcal{E}_{\mathcal{U}}} \left( z_{ij} \hat{y}_{ij} + (1 - z_{ij})(1 - \hat{y}_{ij}) \right) \quad (9)$$

where $\hat{y}_{ij} = f_{\theta^{\text{new}}}(i, j)$, and $z_{ij} \in \{1, 0\}$ represents the connectivity between nodes $i$ and $j$.

Using Equations (2) and (9), we derive Theorem 1 that shows the similarity between the iterative minimization of $\mathcal{L}_E$ in PULL and the iterative maximization of $Q(\theta^{\text{new}} \mid \theta)$ in the EM algorithm.

**Theorem 1.** *Given the assumption in Equation (9), the expected log likelihood $Q(\theta^{\text{new}} \mid \theta)$ of the EM algorithm reduces to the negative of the loss function $\mathcal{L}_E$ of PULL with the expected graph $\bar{\mathcal{G}}$:*

$$Q(\theta^{\text{new}} \mid \theta) \approx \sum_{e_{ij} \in \mathcal{E}_{\mathcal{P}}} \log \hat{y}_{ij} + \sum_{e_{ij} \in \mathcal{E}_{\mathcal{U}}} \left( \mathbf{A}_{ij}^{\bar{\mathcal{G}}} \log \hat{y}_{ij} + (1 - \mathbf{A}_{ij}^{\bar{\mathcal{G}}}) \log(1 - \hat{y}_{ij}) \right) \quad (10)$$

*where $\hat{y}_{ij}$ is the estimated linking probability between nodes $i$ and $j$ by $f_{\theta^{\text{new}}}$, and $\mathbf{A}^{\bar{\mathcal{G}}}$ is the corresponding adjacency matrix of $\bar{\mathcal{G}}$ (see Appendix D for proof).*

## 4 EXPERIMENTS

We conduct diverse experiments on real-world datasets to provide answers to the following questions.

**Q1 Link prediction performance (Section 4.2).** How accurate is PULL compared to the baselines for predicting links in edge-incomplete graphs?

**Q2 Effect of iterative learning (Section 4.3)** How does the accuracy change over iterations?

**Q3 Effect of additional loss (Section 4.4).** How does the additional loss term $\mathcal{L}_C$ of PULL contribute to the performance?

**Q4 Scalability (Section 4.5).** How does the runtime of PULL change as the graph size grows?

### 4.1 EXPERIMENTAL SETTINGS

**Datasets.** We use five real-world datasets from various domains which are summarized in Table 3 (Appendix B.2). PubMed and Cora-full are citation networks where nodes correspond to scientific publications and edges denote citation between the papers. Each node has binary bag-of-words features indicating the presence or absence of specific words from a predefined dictionary. Chameleon and Crocodile are Wikipedia networks, with nodes representing web pages and edges representing hyperlinks between them. Node features include keywords or informative nouns extracted from the Wikipedia pages. Facebook are social networks where each node represents a user, and edges indicate follower relationships. Node features represent user-specific information such as age and gender.

**Baselines.** We compare PULL with previous approaches for link prediction in graphs. GCN+CE, GAT+CE, and SAGE+CE use GCN (Kipf & Welling, 2017), GAT (Velickovic et al., 2017), and GraphSAGE (Hamilton et al., 2017) for computing linking probabilities, respectively. They utilize cross entropy (CE) for training, while randomly sampling $|\mathcal{E}_\mathcal{P}|$ non-edges from $\mathcal{E}_\mathcal{U}$ for every epoch to balance the ratio between edge and non-edge examples. GAE (Kipf & Welling, 2016b) utilizes an autoencoder to compute the linking probabilities, forcing the predicted graph structure to be similar to the given graph. VGAE (Kipf & Welling, 2016b) exploits a variational autoencoder to learn the embedding of edges based on the given graph structure and node features. ARGA and ARGVA (Pan et al., 2018) improve the performance of GAE and VGAE, respectively by introducing adversarial training strategy. GNAE and VGNAE (Ahn & Kim, 2021) utilize L2-normalization to obtain better node embeddings for isolated nodes. BaggingPU (Gan et al., 2022) classifies node pairs into observed and unobserved, and approximates the linking probabilities using the ratio of observed links. All of them including PULL are implemented in Python.

**Evaluation and Settings.** We evaluate the performance of PULL and the baselines in classifying edges and non-edges correctly. We use AUROC score (AUC score) as the main evaluation metric, but also report the AUPRC score for a thorough evaluation following Kipf & Welling (2017). The models are trained using graphs that lack some edges, while preserving all node attributes. The validation and test sets consist of the missing edges and an equal number of randomly sampled non-edges. We vary the ratio $r_m$ of test missing edges in {0.1, 0.2}. The ratio of valid missing edges are set to 0.1 through the experiments. The validation set is used for early stopping with patience 200 while the number of maximum epochs is set to 2,000. For PULL, we set the number of inner loops for training a link predictor $f$ as 200, and the number of iterations as 10. We use Adam optimizer with a learning rate of 0.01, and set the numbers of layers and hidden dimensions as 2 and 16, respectively, following Kipf & Welling (2017) for fair comparison between the methods. For ARGA and ARGVA which utilize adversarial training strategy, we use the default settings described in the paper. We conduct experiments ten times with different random seeds, and present the results in terms of both the average and the standard deviation.

### 4.2 LINK PREDICTION PERFORMANCE (Q1)

We compare the link prediction performance of PULL with the baselines for various ratio $r_m$ of missing edges in Table 1. Note that PULL achieves the highest AUROC and AUPRC scores among the methods in most of the cases. Furthermore, PULL presents the lowest standard deviation compared to the baselines. This highlights the significance of considering the uncertainty of the provided graph structure during the training of the link predictor $f$ to enhance prediction performance.

Table 1: The link prediction accuracy of PULL and baselines in terms of AUROC and AUPRC. Bold numbers denote the best performance, and underlined ones represent the second-best accuracy. Note that PULL outperforms all the baselines in most of the cases.

Missing ratio $r_m = 0.1$

| Model | PubMed | | Cora-full | | Chameleon | | Crocodile | | Facebook | |
|---|---|---|---|---|---|---|---|---|---|---|
| | AUROC | AUPRC | AUROC | AUPRC | AUROC | AUPRC | AUROC | AUPRC | AUROC | AUPRC |
| GCN+CE | $96.4 \pm 0.3$ | $96.5 \pm 0.3$ | $95.6 \pm 0.5$ | $95.2 \pm 0.8$ | $96.5 \pm 0.3$ | $96.1 \pm 0.2$ | $95.9 \pm 0.6$ | $95.5 \pm 1.6$ | $96.6 \pm 0.2$ | $96.9 \pm 0.1$ |
| GAT+CE | $89.9 \pm 0.5$ | $89.4 \pm 0.8$ | $94.1 \pm 0.5$ | $93.4 \pm 0.4$ | $90.3 \pm 0.1$ | $90.2 \pm 0.3$ | $90.2 \pm 0.2$ | $92.4 \pm 0.1$ | $92.1 \pm 0.5$ | $92.1 \pm 0.7$ |
| SAGE+CE | $86.3 \pm 0.7$ | $88.1 \pm 0.2$ | $94.5 \pm 0.5$ | $94.9 \pm 0.5$ | $96.0 \pm 0.7$ | $94.9 \pm 0.1$ | $95.1 \pm 0.7$ | $95.8 \pm 0.4$ | $95.1 \pm 0.3$ | $94.8 \pm 0.6$ |
| GAE | $96.3 \pm 0.2$ | $96.5 \pm 0.1$ | $95.7 \pm 0.7$ | $95.2 \pm 0.9$ | $96.5 \pm 0.3$ | $96.3 \pm 0.3$ | $95.9 \pm 0.7$ | $96.2 \pm 0.5$ | $96.6 \pm 0.2$ | $97.0 \pm 0.2$ |
| VGAE | $94.4 \pm 0.9$ | $93.9 \pm 1.0$ | $93.0 \pm 3.0$ | $88.8 \pm 6.4$ | $96.1 \pm 0.3$ | $96.0 \pm 0.2$ | $94.9 \pm 0.4$ | $93.9 \pm 0.1$ | $93.9 \pm 1.5$ | $95.2 \pm 0.4$ |
| ARGA | $93.6 \pm 0.3$ | $93.5 \pm 0.1$ | $91.3 \pm 0.7$ | $91.1 \pm 0.3$ | $94.9 \pm 0.7$ | $94.5 \pm 0.2$ | $96.1 \pm 0.3$ | $95.7 \pm 0.4$ | $92.0 \pm 0.8$ | $92.2 \pm 0.4$ |
| ARGVA | $93.9 \pm 1.1$ | $94.2 \pm 0.3$ | $90.9 \pm 1.6$ | $89.5 \pm 1.7$ | $93.8 \pm 0.9$ | $93.9 \pm 0.2$ | $95.0 \pm 0.2$ | $94.4 \pm 0.4$ | $92.7 \pm 2.4$ | $89.4 \pm 2.8$ |
| GNAE | $96.0 \pm 0.4$ | $95.8 \pm 0.6$ | $95.8 \pm 0.6$ | $94.9 \pm 0.8$ | $97.8 \pm 0.1$ | $97.6 \pm 0.1$ | $97.7 \pm 0.3$ | $97.7 \pm 0.2$ | $96.0 \pm 0.2$ | $96.2 \pm 0.1$ |
| VGNAE | $94.3 \pm 1.4$ | $94.2 \pm 0.6$ | $93.2 \pm 1.8$ | $93.1 \pm 0.5$ | $96.2 \pm 1.0$ | $96.2 \pm 1.0$ | $94.6 \pm 1.0$ | $92.6 \pm 0.4$ | $93.8 \pm 1.1$ | $95.0 \pm 0.3$ |
| Bagging-PU | $94.6 \pm 0.4$ | $94.9 \pm 0.4$ | $92.6 \pm 0.6$ | $93.9 \pm 0.5$ | $97.4 \pm 0.6$ | $97.4 \pm 0.7$ | $97.4 \pm 0.3$ | $97.7 \pm 0.3$ | $97.0 \pm 0.1$ | $97.4 \pm 0.1$ |
| PULL (ours) | $96.6 \pm 0.2$ | $96.9 \pm 0.1$ | $96.1 \pm 0.3$ | $96.4 \pm 0.4$ | $97.9 \pm 0.2$ | $97.9 \pm 0.2$ | $98.3 \pm 0.1$ | $98.5 \pm 0.1$ | $97.4 \pm 0.1$ | $97.7 \pm 0.1$ |

Missing ratio $r_m = 0.2$

| Model | PubMed | | Cora-full | | Chameleon | | Crocodile | | Facebook | |
|---|---|---|---|---|---|---|---|---|---|---|
| | AUROC | AUPRC | AUROC | AUPRC | AUROC | AUPRC | AUROC | AUPRC | AUROC | AUPRC |
| GCN+CE | $96.0 \pm 0.2$ | $96.0 \pm 0.3$ | $95.1 \pm 0.7$ | $95.2 \pm 0.5$ | $96.4 \pm 0.2$ | $96.2 \pm 0.1$ | $96.1 \pm 0.6$ | $96.5 \pm 0.7$ | $96.5 \pm 0.3$ | $96.8 \pm 0.2$ |
| GAT+CE | $89.7 \pm 0.2$ | $89.5 \pm 0.4$ | $94.1 \pm 0.3$ | $93.6 \pm 0.2$ | $90.7 \pm 0.1$ | $90.7 \pm 0.7$ | $91.0 \pm 0.1$ | $92.1 \pm 0.2$ | $91.9 \pm 0.5$ | $92.0 \pm 0.2$ |
| SAGE+CE | $84.8 \pm 0.5$ | $87.3 \pm 0.8$ | $93.8 \pm 0.7$ | $94.7 \pm 0.3$ | $95.8 \pm 0.5$ | $95.0 \pm 0.4$ | $95.0 \pm 0.7$ | $95.7 \pm 0.8$ | $94.7 \pm 0.4$ | $94.8 \pm 0.3$ |
| GAE | $96.0 \pm 0.1$ | $95.9 \pm 0.2$ | $95.1 \pm 0.5$ | $95.1 \pm 0.4$ | $96.4 \pm 0.2$ | $96.3 \pm 0.2$ | $95.8 \pm 0.6$ | $96.2 \pm 0.6$ | $96.6 \pm 0.2$ | $96.9 \pm 0.1$ |
| VGAE | $93.8 \pm 1.3$ | $93.7 \pm 1.1$ | $91.5 \pm 3.0$ | $88.1 \pm 5.3$ | $95.9 \pm 0.6$ | $95.9 \pm 0.2$ | $94.7 \pm 0.3$ | $94.3 \pm 0.9$ | $94.3 \pm 0.9$ | $94.5 \pm 0.4$ |
| ARGA | $93.2 \pm 0.7$ | $93.1 \pm 0.4$ | $90.4 \pm 1.0$ | $90.2 \pm 0.5$ | $94.8 \pm 0.4$ | $94.7 \pm 0.3$ | $95.9 \pm 0.5$ | $95.5 \pm 0.6$ | $91.8 \pm 0.7$ | $91.8 \pm 0.2$ |
| ARGVA | $93.5 \pm 1.2$ | $93.4 \pm 0.4$ | $88.2 \pm 3.8$ | $84.3 \pm 4.3$ | $93.6 \pm 0.5$ | $93.8 \pm 0.2$ | $94.9 \pm 0.3$ | $94.1 \pm 0.1$ | $92.5 \pm 2.4$ | $92.7 \pm 1.8$ |
| GNAE | $95.7 \pm 0.3$ | $95.6 \pm 0.3$ | $95.5 \pm 0.7$ | $94.8 \pm 1.0$ | $97.7 \pm 0.3$ | $97.5 \pm 0.1$ | $97.6 \pm 0.2$ | $97.7 \pm 0.2$ | $95.8 \pm 0.3$ | $96.1 \pm 0.1$ |
| VGNAE | $93.3 \pm 1.2$ | $93.7 \pm 1.1$ | $92.5 \pm 2.1$ | $93.8 \pm 0.5$ | $95.6 \pm 0.6$ | $95.4 \pm 0.8$ | $94.6 \pm 0.9$ | $91.7 \pm 1.4$ | $93.7 \pm 1.0$ | $94.1 \pm 1.2$ |
| Bagging-PU | $94.0 \pm 0.3$ | $94.4 \pm 0.4$ | $92.3 \pm 0.7$ | $94.2 \pm 0.6$ | $97.4 \pm 0.3$ | $97.4 \pm 0.3$ | $97.5 \pm 0.4$ | $97.8 \pm 0.4$ | $96.9 \pm 0.2$ | $97.2 \pm 0.1$ |
| PULL (ours) | $96.3 \pm 0.1$ | $96.5 \pm 0.1$ | $95.4 \pm 0.3$ | $95.7 \pm 0.4$ | $97.9 \pm 0.1$ | $97.7 \pm 0.2$ | $98.3 \pm 0.1$ | $98.4 \pm 0.1$ | $97.4 \pm 0.1$ | $97.6 \pm 0.1$ |

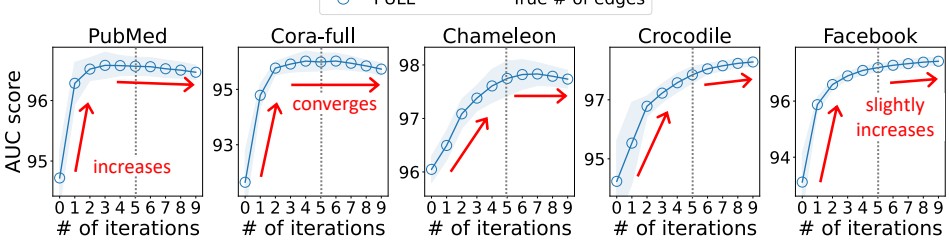

Figure 2: AUC score of PULL through the iterations. The gray dashed lines denote the ground-truth numbers of edges. The performance of PULL increases as the iteration proceeds. The accuracy converges or slightly increases as the number $K$ of sampled edges exceeds the ground-truth one.

It is also noteworthy that GCN+CE model, which propagates information through the edge-incomplete graph using GCN, shows consistently lower performance than PULL. This shows that the propagation of PULL with the expected graph structure effectively prevents $f$ from overfitting to the given graph structure, whereas the propagation of GCN+CE with the given graph leads to overfitting.

## 4.3 EFFECT OF ITERATIVE LEARNING (Q2)

For each iteration, PULL computes the expected graph $\bar{\mathcal{G}}$ utilizing the trained link predictor $f$ from the previous iteration. Then PULL retrains $f$ with $\bar{\mathcal{G}}$ to prevent the link predictor from overfitting to the given graph, thus enhancing the link prediction performance. We study how the prediction accuracy of PULL evolves as the iteration proceeds in Figure 2. PULL increases the number $K$ of selected edges for the approximation of $\bar{\mathcal{G}}$ as the iteration progresses. The gray dashed lines indicate the points at which $K$ becomes equal to the ground-truth number of edges for each dataset.

The AUC score of PULL in Figure 2 increases through the iterations, reaching its best performance when the number $K$ of selected edges closely matches the ground-truth one. This shows that PULL enhances the quality of the expected graph as the iterations progress, and eventually makes accurate predictions of the true graph structure. In PubMed, Cora-full, and Chameleon, the accuracy converges or slightly decreases when the number $K$ exceeds that of ground-truth edges. This is due to the oversmoothing problem caused by propagating information through a graph with more edges than the

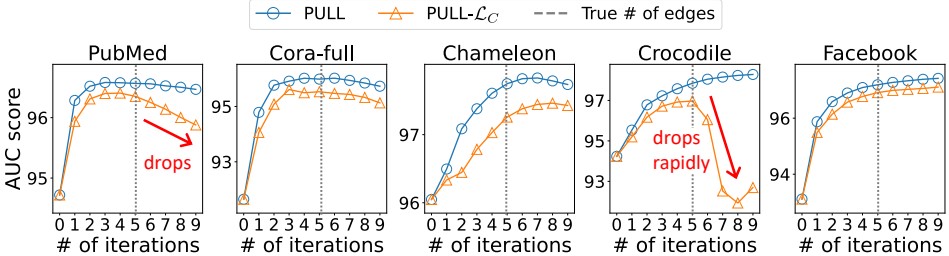

Figure 4: The effect of $\mathcal{L}_C$ on the link prediction performance of PULL. PULL-$\mathcal{L}_C$ represents PULL without $\mathcal{L}_C$. PULL consistently shows superior performance than PULL-$\mathcal{L}_C$. In PubMed and Crocodile, the accuracy of PULL-$\mathcal{L}_C$ drops rapidly after exceeding the gray dashed lines which indicate the true number of edges.

true graph. In Crocodile and Facebook, the prediction accuracy increases even with larger number of edges than the ground-truth one. This observation indicates that both the ground-truth graph structures of Crocodile and Facebook inherently contain missing links.

## 4.4 EFFECT OF ADDITIONAL LOSS (Q3)

We study the effect of the additional loss term $\mathcal{L}_C$ of PULL on the link prediction performance. We report the AUC scores through the iterations in Figure 4. PULL-$\mathcal{L}_C$ represents PULL trained by minimizing only $\mathcal{L}_E$. Note that PULL-$\mathcal{L}_C$ consistently shows lower prediction accuracy than PULL. In PubMed and Crocodile, the AUC scores of PULL-$\mathcal{L}_C$ drop rapidly after the fifth iteration, where the number $K$ of selected edges exceeds the ground-truth one. This indicates that $\mathcal{L}_C$ effectively safeguards PULL against performance degradation when the expected graph structure contains more number of edges than the actual one.

## 4.5 SCALABILITY (Q4)

We investigate the running time of PULL on subgraphs with different sizes to show its scalability to large graphs in Figure 3. To generate the subgraphs, we randomly sample edges from the original graphs with various portions $r_p \in \{0.1, ..., 0.9\}$. Thus, each induced subgraph has $r_p|\mathcal{E}|$ edges where $\mathcal{E}$ is the set of edges of the original graph. Figure 3 shows that the running time of PULL exhibits a linear increase with the number of edges, showing its scalability to large graphs. This is because PULL effectively approximates the expected graph $\bar{\mathcal{G}}$ with $|\mathcal{V}|^2$ weighted edges to a graph $\bar{\mathcal{G}}'$ with $(1 + 0.05t)|\mathcal{E}_\mathcal{P}|$ edges where $\mathcal{V}$ and $\mathcal{E}_\mathcal{P}$ are sets of nodes and observed edges, respectively, and $t$ is the number of iterations.

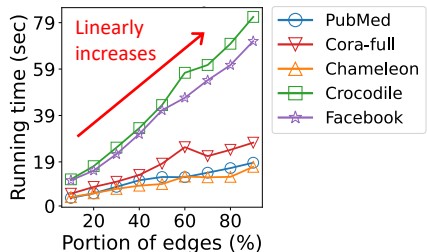

Figure 3: Runtime of PULL on sampled subgraphs. The time increases linearly with the size growth.

## 5 CONCLUSION

We propose PULL, an accurate method for link prediction in edge-incomplete graphs. PULL addresses the limitation of previous approaches, which is their heavy reliance on the observed graph, by iteratively predicting the true graph structure. PULL proposes latent variables for the unconnected edges in a graph, and propagate information through the expected graph structure. PULL then uses the expected linking probabilities of unconnected edges as their pseudo labels for training a link predictor. Extensive experiments on five real-worlds datasets show that PULL shows superior performance than the baselines. A potential drawback of PULL is that changing labels and graph structure might require additional training time for the link predictor to adapt to these modifications. Furthermore, it is not easy for PULL to be directly applied to multi-relational graphs since PULL predict links based on the similarity between nodes. Future work includes streamlining the iterative improvement of the expected graph structure into a single iteration, and extending PULL to multi-relational graphs that incorporate richer relationships, such as like or hate between nodes.

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

## A   PROBLEM DEFINITION

We formally define the problem of link prediction in Problem 1.

**Problem 1** (Link Prediction in Edge-incomplete Graphs). *We have an edge-incomplete graph $\mathcal{G}_{\mathcal{P}} = (\mathcal{V}, \mathcal{E}_{\mathcal{P}})$, along with a feature matrix $\mathbf{X} \in \mathbb{R}^{|\mathcal{V}| \times d}$ where $\mathcal{V}$ and $\mathcal{E}_{\mathcal{P}}$ are the sets of nodes and observed edges, respectively, and $d$ is the number of features for each node. The remaining unconnected node pairs are denoted as a set $\mathcal{E}_{\mathcal{U}}$. The objective of link prediction in edge-incomplete graphs is to train a link predictor $f$ that accurately identifies the connected node pairs within $\mathcal{E}_{\mathcal{U}}$.*

## B   SYMBOLS AND DATASETS

### B.1   SYMBOLS

Table 2: Symbols.

| Symbol | Description |
|---|---|
| $\mathcal{G}_{\mathcal{P}} = (\mathcal{V}, \mathcal{E}_{\mathcal{P}})$ | Edge incomplete graph with sets $\mathcal{V}$ of nodes and $\mathcal{E}_{\mathcal{P}}$ of observed edges |
| $\mathcal{E}_{\mathcal{U}}$ | Set of unconnected node pairs (unconnected edges) |
| $e_{ij}$ | Edge between nodes $i$ and $j$ |
| $L(\mathcal{G})$ | Corresponding line graph of $\mathcal{G}$ |
| $\mathbf{A}^{\mathcal{G}}$ | Corresponding adjacency matrix of $\mathcal{G} = (\mathcal{V}, \mathcal{E})$ where $\mathbf{A}_{ij}^{\mathcal{G}} = 1$ if $e_{ij} \in \mathcal{E}$ |
| $\mathbf{X}$ | Feature matrix for every node in $\mathcal{G}_{\mathcal{P}}$ |
| $f_{\theta}(\cdot, \cdot)$ | Link predictor parameterized by $\theta$ |
| $\hat{y}_{ij}$ | $f_{\theta}(i, j)$ representing the marginal linking probabilities between nodes $i$ and $j$ |
| $\mathcal{L}(\cdot)$ | Objective function that PULL aims to minimize |

### B.2   DATASETS

Table 3: Summary of datasets.

| Datasets | # of nodes | # of edges | # of features | Description |
|---|---|---|---|---|
| PubMed[1] | 19,717 | 88,648 | 500 | Citation |
| Cora-full[2] | 19,793 | 126,842 | 8,710 | Citation |
| Chameleon[3] | 2,277 | 36,101 | 2,325 | Wikipedia |
| Crocodile[3] | 11,631 | 191,506 | 500 | Wikipedia |
| Facebook[4] | 22,470 | 342,004 | 128 | Social |
| Physics[5] | 34,493 | 495,924 | 8,415 | Citation |
| ogbn-arxiv[6] | 169,343 | 1,166,243 | 128 | Citation |

[1] https://github.com/kimiyoung/planetoid
[2] https://www.cs.cit.tum.de/daml/g2g/
[3] https://snap.stanford.edu/data/wikipedia-article-networks.html
[4] https://github.com/benedekrozemberczki/MUSAE
[5] https://github.com/shchur/gnn-benchmark/raw/master/data/npz/
[6] https://ogb.stanford.edu/docs/nodeprop/#ogbn-arxiv

## C  LEMMA

**Lemma 1.** *We are given a graph $\mathcal{G}_\mathcal{P}$ and its corresponding line graph $L(\mathcal{G}_\mathcal{P}) = (\mathcal{V}_L, \mathcal{E}_L)$ where $\mathcal{V}_L$ and $\mathcal{E}_L$ are sets of nodes and edges in $L(\mathcal{G}_\mathcal{P})$, respectively. We are also given node potentials $\phi_{ij}(z_{ij} \mid \theta)$ of nodes $ij$ in graph $L(\mathcal{G}_\mathcal{P})$. Then the following equation holds for $\sum_{z_{ij}} \phi_{ij}(z_{ij} \mid \theta) = 1$:*

$$\sum_{\mathbf{z}} \prod_{ij \in \mathcal{V}_L} \phi_{ij}(z_{ij} \mid \theta) = 1. \tag{11}$$

*Proof.* Let $N = |\mathcal{V}_L|$, and $\mathcal{E}$ be the set of all observed edges and unconnected edges in $\mathcal{G}_\mathcal{P}$. Then the sum of $\prod_{ij \in \mathcal{V}_L} \phi_{ij}(z_{ij}|\theta)$ for all possible $\mathbf{z}$ is computed as follows:

$$\begin{aligned}
\sum_{\mathbf{z}} \prod_{ij \in \mathcal{V}_L} \phi_{ij}(z_{ij} \mid \theta) &= \sum_{\mathbf{z}} \prod_{e_{ij} \in \mathcal{E}} \phi_{ij}(z_{ij} \mid \theta) \\
&= \sum_{\mathbf{z} \setminus \{z_{11}\}} \prod_{e_{ij} \in \mathcal{E} \setminus \{e_{11}\}} \phi_{ij}(z_{ij} \mid \theta) \sum_{z_{11}} \phi_{11}(z_{11} \mid \theta) \\
&= \sum_{\mathbf{z} \setminus \{z_{11}, z_{12}\}} \prod_{e_{ij} \in \mathcal{E} \setminus \{e_{11}, e_{12}\}} \phi_{ij}(z_{ij} \mid \theta) \sum_{z_{12}} \phi_{12}(z_{12} \mid \theta) \\
&\quad\quad\quad\quad \vdots \\
&= \sum_{z_{NN}} \phi_{NN}(z_{NN} \mid \theta) = 1
\end{aligned} \tag{12}$$

which ends the proof. Similarly, we prove that $\sum_{\mathbf{z}|z_{ij}=1} \prod_{e_{kl} \in \mathcal{E}_\mathcal{U} \setminus \{e_{ij}\}} \phi_{kl}(z_{kl} \mid \theta) = 1$.  □

## D  PROOF OF THEOREM 1

*Proof.* Using Equations (2) and (9), the expected log likelihood $Q(\theta^{\text{new}} \mid \theta)$ is expressed as follows:

$$\begin{aligned}
Q(\theta^{\text{new}} \mid \theta) &= \sum_{\mathbf{z}} p(\mathbf{z} \mid \mathbf{X}, \mathcal{E}_\mathcal{P}, \theta) \log p(\mathcal{E}_\mathcal{P}, \mathbf{z} \mid \mathbf{X}, \theta^{\text{new}}) \\
&\approx \sum_{\mathbf{z}} p(\mathbf{z} \mid \mathbf{X}, \mathcal{E}_\mathcal{P}, \theta) \Big( \sum_{e_{ij} \in \mathcal{E}_\mathcal{P}} \log \hat{y}_{ij} + \sum_{e_{ij} \in \mathcal{E}_\mathcal{U}} \log \big( z_{ij} \hat{y}_{ij} + (1 - z_{ij})(1 - \hat{y}_{ij}) \big) \Big) \\
&= \sum_{e_{ij} \in \mathcal{E}_\mathcal{P}} \log \hat{y}_{ij} + \sum_{\mathbf{z}} \prod_{e_{kl} \in \mathcal{E}_\mathcal{U}} \phi_{kl}(z_{kl} \mid \theta) \sum_{e_{ij} \in \mathcal{E}_\mathcal{U}} \log \big( z_{ij} \hat{y}_{ij} + (1 - z_{ij})(1 - \hat{y}_{ij}) \big).
\end{aligned} \tag{13}$$

The last term $\sum_{\mathbf{z}} \prod_{e_{kl} \in \mathcal{E}_\mathcal{U}} \phi_{kl}(z_{kl} \mid \theta) \sum_{e_{ij} \in \mathcal{E}_\mathcal{U}} \log \big( z_{ij} \hat{y}_{ij} + (1 - z_{ij})(1 - \hat{y}_{ij}) \big)$ in Equation (13) is expressed as follows:

$$\begin{aligned}
&\sum_{\mathbf{z}} \prod_{e_{kl} \in \mathcal{E}_\mathcal{U}} \phi_{kl}(z_{kl} \mid \theta) \sum_{e_{ij} \in \mathcal{E}_\mathcal{U}} \log \big( z_{ij} \hat{y}_{ij} + (1 - z_{ij})(1 - \hat{y}_{ij}) \big) \\
&= \sum_{\mathbf{z}} \prod_{e_{kl} \in \mathcal{E}_\mathcal{U} \setminus \{e_{ij}\}} \phi_{kl}(z_{kl} \mid \theta) \sum_{e_{ij} \in \mathcal{E}_\mathcal{U}} \phi_{ij}(z_{ij} \mid \theta) \log \big( z_{ij} \hat{y}_{ij} + (1 - z_{ij})(1 - \hat{y}_{ij}) \big) \\
&= \sum_{e_{ij} \in \mathcal{E}_\mathcal{U}} \big( \phi_{ij}(z_{ij} = 1 \mid \theta) \log \hat{y}_{ij} + \phi_{ij}(z_{ij} = 0 \mid \theta) \log(1 - \hat{y}_{ij}) \big) \\
&= \sum_{e_{ij} \in \mathcal{E}_\mathcal{U}} \big( \mathbf{A}_{ij}^{\bar{\mathcal{G}}} \log \hat{y}_{ij} + (1 - \mathbf{A}_{ij}^{\bar{\mathcal{G}}}) \log(1 - \hat{y}_{ij}) \big)
\end{aligned} \tag{14}$$

where the second equality uses the fact that $\sum_{\mathbf{z} \setminus \{z_{ij}\}} \prod_{e_{kl} \in \mathcal{E}_\mathcal{U} \setminus \{e_{ij}\}} \phi_{kl}(z_{kl} \mid \theta) = 1$ (from Lemma 1), and the third equality uses Equation (4).

Using the result of Equation (14), the expected log likelihood $Q(\theta^{\text{new}} \mid \theta)$ in Equation (13) reduces to the negative of the loss function $\mathcal{L}_E$ of PULL:

$$Q(\theta^{\text{new}} \mid \theta) \approx \sum_{e_{ij} \in \mathcal{E}_\mathcal{P}} \log \hat{y}_{ij} + \sum_{e_{ij} \in \mathcal{E}_\mathcal{U}} \big( \mathbf{A}_{ij}^{\bar{\mathcal{G}}} \log \hat{y}_{ij} + (1 - \mathbf{A}_{ij}^{\bar{\mathcal{G}}}) \log(1 - \hat{y}_{ij}) \big) \tag{15}$$

which ends the proof. Note that Equation (15) uses $\bar{\mathcal{G}}$ which is approximated to $\bar{\mathcal{G}}'$ in PULL.  □

# E   DETAILED SETTINGS OF EXPERIMENTS

We provide detailed settings of hyperparameters for PULL and the baselines, which are not presented in Section 4.1. All the experiments are conducted under a single GPU machine with GTX 1080 Ti.

**GCN+CE.** We use GCN code[1] implemented with torch-geometric package. For each epoch, the model randomly samples $|\mathcal{E}_{\mathcal{P}}|$ negative samples (unconnected node pairs), and minimizes the cross entropy loss.

**GAT+CE.** We use GAT code[1] implemented with torch-geometric package. For each epoch, GAT+CE randomly samples $|\mathcal{E}_{\mathcal{P}}|$ negative samples, and minimizes the cross entropy loss. We set the multi-head attention number as 8 with mean aggregation strategy, and the dropout ratio as 0.6 following the original paper (Velickovic et al., 2017).

**SAGE+CE.** We use GraphSAGE code[1] implemented with torch-geometric package. For each epoch, the model randomly samples $|\mathcal{E}_{\mathcal{P}}|$ negative samples, and minimizes the cross entropy loss. We use mean aggregation scheme following the original paper (Hamilton et al., 2017).

**GAE & VGAE.** We use GAE and VGAE codes[2] implemented with torch-geometric package. We use GCN-based encoder and decoder for both GAE and VGAE following the original paper (Kipf & Welling, 2016b). The number of layers and units for decoders are set to 2 and 16, respectively.

**ARGA & ARGVA.** We use ARGA and ARGVA codes[2] implemented with torch-geometric package. We use the same hyperparameter settings for the adversarial training of them as presented in the original paper (Pan et al., 2018).

**GNAE & VGNAE.** We use GNAE and VGNAE codes[3] implemented by the authors. The scaling constant $s$ is set to 1.8 following the original paper (Ahn & Kim, 2021).

**Bagging-PU.** We reimplement Bagging-PU since there is no public implementation of authors. We use GCN instead of SDNE (Wang et al., 2016b) for the node embedding model since SDNE is an unsupervised representation-based method, which limits the performance. We use the mean aggregation following the original paper (Gan et al., 2022), and set the bagging size as three.

**PULL.** We use torch-geometric (Fey & Lenssen, 2019) package to implement weighted propagation of GCN. The number of inner epochs is set to 200, while that of outer iteration is set to 10. We increase the number $K$ of edges in the approximated version of expected graph $\bar{\mathcal{G}}$ in proportion to that of observed edges through the iterations: $K \leftarrow K + r|\mathcal{E}_{\mathcal{P}}|$ where $r$ is the increasing ratio. We set $r = 0.05$ in our experiments. The code and data for PULL are available at https://github.com/graphmaster2023/pull.

---

[1]https://github.com/pyg-team/pytorch_geometric/blob/master/torch_geometric/nn/models/basic_gnn.py
[2]https://github.com/pyg-team/pytorch_geometric/blob/master/torch_geometric/nn/models/autoencoder.py
[3]https://github.com/SeongJinAhn/VGNAE

# F   FURTHER EXPERIMENTS

## F.1   APPLYING PULL TO OTHER GCN-BASED METHODS

PULL can be applied to other GCN-based methods including GAE, VGAE, ARGA, and ARGVA that use GCN-based propagation scheme. It is not easy for PULL to be directly applied to other baselines such as GAT, GraphSAGE, GNAE and VGNAE. This is because they use different propagation scheme instead of GCN, posing a challenge for PULL in propagating information through the expected graph structure during training. For example, GAT propagates information only through the observed edges using the attention scores as weights. GraphSAGE performs random walks to define adjacent nodes. GNAE and VGNAE separate the feature mapping and propagation processes.

To demonstrate that PULL improves the performance of existing models, we conduct experiments by applying our method to GAE, VGAE, ARGA, and ARGVA. We conduct experiments three times with random seeds, while using the same experimental settings as in Section 4.1. Table 4 summarizes the results. Note that PULL improves the performance of the baselines in most of the cases, showing its applicability across various models.

Table 4: The performance improvement of baselines with the integration of PULL. The best performance is indicated in bold. Note that PULL enhances the performance of baseline models in most cases, demonstrating its effectiveness across a range of different models.

| Missing ratio $r_m = 0.1$ | | | | | | | | | | |
|---|---|---|---|---|---|---|---|---|---|---|
| **Model** | **PubMed** | | **Cora-full** | | **Chameleon** | | **Crocodile** | | **Facebook** | |
| | AUROC | AUPRC | AUROC | AUPRC | AUROC | AUPRC | AUROC | AUPRC | AUROC | AUPRC |
| GAE | 96.0 ± 0.1 | 95.9 ± 0.2 | 95.1 ± 0.5 | 95.1 ± 0.4 | 96.4 ± 0.2 | 96.3 ± 0.2 | 95.8 ± 0.6 | 96.2 ± 0.6 | 96.6 ± 0.2 | 96.9 ± 0.1 |
| GAE+PULL | **96.5 ± 0.1** | **96.7 ± 0.2** | **95.4 ± 0.5** | **95.6 ± 0.5** | **97.8 ± 0.2** | **97.8 ± 0.2** | **97.5 ± 0.4** | **97.3 ± 0.4** | **97.2 ± 0.3** | **97.5 ± 0.2** |
| VGAE | 93.8 ± 1.3 | 93.7 ± 1.1 | 91.5 ± 3.0 | 88.1 ± 5.3 | 95.9 ± 0.6 | 95.9 ± 0.2 | 94.7 ± 0.3 | 94.3 ± 0.9 | 94.3 ± 0.9 | 94.5 ± 0.4 |
| VGAE+PULL | **95.2 ± 0.4** | **95.2 ± 0.3** | **93.8 ± 0.3** | **93.9 ± 0.4** | **97.0 ± 0.0** | **97.1 ± 0.1** | **96.3 ± 0.2** | **96.3 ± 0.2** | **96.4 ± 0.3** | **96.6 ± 0.3** |
| ARGA | 93.2 ± 0.7 | 93.1 ± 0.4 | 90.4 ± 1.0 | 90.2 ± 0.5 | 94.8 ± 0.4 | 94.7 ± 0.3 | 95.9 ± 0.5 | 95.5 ± 0.6 | 91.8 ± 0.7 | 91.8 ± 0.2 |
| ARGA+PULL | **93.9 ± 0.6** | **94.6 ± 0.5** | **94.4 ± 1.0** | **93.9 ± 1.2** | **96.3 ± 0.3** | **96.0 ± 0.3** | **95.2 ± 0.6** | **95.7 ± 0.7** | **93.5 ± 0.6** | **93.8 ± 0.6** |
| ARGVA | 93.5 ± 1.2 | 93.4 ± 0.4 | 88.2 ± 3.8 | 84.3 ± 4.3 | 93.6 ± 0.5 | 93.8 ± 0.2 | 94.9 ± 0.3 | 94.1 ± 0.1 | 92.5 ± 2.4 | 92.7 ± 1.8 |
| ARGVA+PULL | **94.8 ± 0.3** | **94.9 ± 0.4** | **93.7 ± 0.4** | **93.6 ± 0.5** | **95.3 ± 0.3** | **95.1 ± 0.0** | **95.4 ± 0.3** | **95.5 ± 0.3** | **94.8 ± 0.2** | **95.2 ± 0.1** |

| Missing ratio $r_m = 0.2$ | | | | | | | | | | |
|---|---|---|---|---|---|---|---|---|---|---|
| **Model** | **PubMed** | | **Cora-full** | | **Chameleon** | | **Crocodile** | | **Facebook** | |
| | AUROC | AUPRC | AUROC | AUPRC | AUROC | AUPRC | AUROC | AUPRC | AUROC | AUPRC |
| GAE | 96.0 ± 0.1 | 95.9 ± 0.2 | **95.1 ± 0.5** | 95.1 ± 0.4 | 96.4 ± 0.2 | 96.3 ± 0.2 | 95.8 ± 0.6 | 96.2 ± 0.6 | 96.6 ± 0.2 | 96.9 ± 0.1 |
| GAE+PULL | **96.1 ± 0.1** | **96.2 ± 0.1** | 94.9 ± 0.4 | **95.2 ± 0.5** | **97.9 ± 0.2** | **97.9 ± 0.1** | **97.5 ± 0.4** | **97.4 ± 0.5** | **97.1 ± 0.2** | **97.3 ± 0.1** |
| VGAE | 93.8 ± 1.3 | 93.7 ± 1.1 | **91.5 ± 3.0** | 88.1 ± 5.3 | 95.9 ± 0.6 | 95.9 ± 0.2 | 94.7 ± 0.3 | 94.3 ± 0.9 | 94.3 ± 0.9 | 94.5 ± 0.4 |
| VGAE+PULL | **94.6 ± 0.5** | **94.5 ± 0.4** | 90.3 ± 5.2 | **90.4 ± 4.8** | **96.9 ± 0.2** | **97.0 ± 0.2** | **96.5 ± 0.1** | **96.5 ± 0.1** | **96.1 ± 0.3** | **96.3 ± 0.2** |
| ARGA | 93.2 ± 0.7 | 93.1 ± 0.4 | 90.4 ± 1.0 | 90.2 ± 0.5 | 94.8 ± 0.4 | 94.7 ± 0.3 | 95.9 ± 0.5 | 95.5 ± 0.6 | 91.8 ± 0.7 | 91.8 ± 0.2 |
| ARGA+PULL | **93.3 ± 0.9** | **93.6 ± 1.0** | **91.6 ± 3.3** | **91.6 ± 3.0** | **96.8 ± 0.2** | **96.8 ± 0.2** | **96.0 ± 0.5** | **95.9 ± 0.4** | **93.5 ± 0.1** | **93.7 ± 0.4** |
| ARGVA | 93.5 ± 1.2 | 93.4 ± 0.4 | 88.2 ± 3.8 | 84.3 ± 4.3 | 93.6 ± 0.5 | 93.8 ± 0.2 | 94.9 ± 0.3 | 94.1 ± 0.1 | 92.5 ± 2.4 | 92.7 ± 1.8 |
| ARGVA+PULL | **94.7 ± 0.1** | **94.8 ± 0.1** | **92.9 ± 1.2** | **92.7 ± 1.3** | **95.9 ± 0.2** | **95.9 ± 0.2** | **95.4 ± 0.4** | **95.9 ± 0.3** | **94.7 ± 0.3** | **95.1 ± 0.3** |

## F.2   WEIGHTED RANDOM SAMPLING FOR CONSTRUCTING $\bar{\mathcal{G}}'$

PULL keeps the top-$K$ edges with highest linking probability to approximate $\bar{\mathcal{G}}$. In this section, we compare PULL with PULL-WS (PULL with Weighted Sampling) that constructs the approximated version $\bar{\mathcal{G}}'$ by performing weighted random sampling of edges from $\bar{\mathcal{G}}$ based on the linking probabilities. As the weighted random sampling empowers PULL to mitigate the excessive self-reinforcement in the link predictor, we additionally exclude the loss term $\mathcal{L}_C$, which serves the same purpose. We conduct experiments three times with random seeds, while using the same experimental settings as in Section 4.1.

Table 5 shows that PULL-WS presents marginal performance decrease compared to PULL. This indicates that keeping the top-$K$ edges with highest linking probability with an additional loss term $\mathcal{L}_C$ shows better link prediction performance than performing weighted random sampling of edges without $\mathcal{L}_C$. However, PULL-WS is an efficient variant of PULL that uses only a single loss term $\mathcal{L}'_E$ instead of the proposed loss $\mathcal{L} = \mathcal{L}'_E + \mathcal{L}_C$.

Table 5: The link prediction accuracy of PULL and its variant PULL-WS. PULL-WS is PULL that approximates $\bar{\mathcal{G}}$ by performing weighted random sampling of edges based on the linking probabilities. Bold numbers denote the highest performance.

| | | | | | | | | | | |
|---|---|---|---|---|---|---|---|---|---|---|
| Missing ratio $r_m = 0.1$ | | | | | | | | | | |
| **Model** | **PubMed** | | **Cora-full** | | **Chameleon** | | **Crocodile** | | **Facebook** | |
| | AUROC | AUPRC | AUROC | AUPRC | AUROC | AUPRC | AUROC | AUPRC | AUROC | AUPRC |
| PULL-WS | $96.5 \pm 0.2$ | $96.7 \pm 0.1$ | $96.0 \pm 0.4$ | $96.3 \pm 0.4$ | $97.4 \pm 0.1$ | $97.5 \pm 0.2$ | $97.6 \pm 0.1$ | $98.0 \pm 0.1$ | $97.2 \pm 0.1$ | $97.6 \pm 0.1$ |
| PULL (ours) | $\mathbf{96.6 \pm 0.2}$ | $\mathbf{96.9 \pm 0.1}$ | $\mathbf{96.1 \pm 0.3}$ | $\mathbf{96.4 \pm 0.4}$ | $\mathbf{97.9 \pm 0.2}$ | $\mathbf{97.9 \pm 0.2}$ | $\mathbf{98.3 \pm 0.1}$ | $\mathbf{98.5 \pm 0.1}$ | $\mathbf{97.4 \pm 0.1}$ | $\mathbf{97.7 \pm 0.1}$ |

| | | | | | | | | | | |
|---|---|---|---|---|---|---|---|---|---|---|
| Missing ratio $r_m = 0.2$ | | | | | | | | | | |
| **Model** | **PubMed** | | **Cora-full** | | **Chameleon** | | **Crocodile** | | **Facebook** | |
| | AUROC | AUPRC | AUROC | AUPRC | AUROC | AUPRC | AUROC | AUPRC | AUROC | AUPRC |
| PULL-WS | $96.2 \pm 0.1$ | $96.4 \pm 0.2$ | $95.2 \pm 0.5$ | $95.7 \pm 0.5$ | $97.8 \pm 0.2$ | $\mathbf{97.8 \pm 0.2}$ | $97.7 \pm 0.1$ | $97.9 \pm 0.1$ | $97.1 \pm 0.0$ | $97.4 \pm 0.1$ |
| PULL (ours) | $\mathbf{96.3 \pm 0.1}$ | $\mathbf{96.5 \pm 0.1}$ | $\mathbf{95.4 \pm 0.3}$ | $\mathbf{95.7 \pm 0.4}$ | $\mathbf{97.9 \pm 0.1}$ | $97.7 \pm 0.2$ | $\mathbf{98.3 \pm 0.1}$ | $\mathbf{98.4 \pm 0.1}$ | $\mathbf{97.4 \pm 0.1}$ | $\mathbf{97.6 \pm 0.1}$ |

## F.3 PERFORMANCE OF PULL IN LARGER NETWORKS

We additionally perform link prediction on two larger graph datasets compared to those discussed in Section 4: ogbn-arxiv (Hu et al., 2020) and Physics[4]. The ogbn-arxiv dataset is a citation network consisting of 169,343 nodes and 1,166,243 edges, where each node represents an arXiv paper and an edge indicates that one paper cites another one. Each node has 128-dimensional feature vector, which is derived by averaging the embeddings of the words in its title and abstract. Physics is a co-authorship graph based on the Microsoft Academic Graph from the KDD Cup 2016 challenge 3. Physics contains 34,493 nodes and 495,924 edges where each node represents an author, and they are connected if they co-authored a paper. For PULL, we set the maximum number of iterations as 20. For the baselines, we set the maximum number of epochs as 4,000. This is because larger data size requires a greater number of epochs to train the link predictor. For other settings, we used the same experimental settings as in Section 4.1. We conduct experiments three times with random seeds.

Table 6 presents the link prediction performance of PULL and the baselines in ogbn-arxiv and Physics. Note that PULL consistently shows superior performance than the baselines in terms of both AUROC and AUPRC. This indicates that PULL is also effective in handling larger graphs.

Table 6: The link prediction accuracy of PULL and the baselines in large graph datasets. Bold numbers denote the highest performance.

| | | | | |
|---|---|---|---|---|
| Missing ratio $r_m = 0.1$ | | | | |
| **Model** | **Physics** | | **ogbn-arxiv** | |
| | AUROC | AUPRC | AUROC | AUPRC |
| GCN+CE | $96.2 \pm 0.0$ | $95.9 \pm 0.1$ | $80.4 \pm 0.3$ | $84.7 \pm 0.6$ |
| GAT+CE | $93.5 \pm 0.1$ | $92.1 \pm 0.1$ | $82.5 \pm 0.2$ | $79.9 \pm 0.5$ |
| SAGE+CE | $95.8 \pm 0.3$ | $95.3 \pm 0.3$ | $82.4 \pm 1.8$ | $80.5 \pm 1.2$ |
| GAE | $\underline{96.2 \pm 0.1}$ | $95.8 \pm 0.1$ | $80.4 \pm 0.3$ | $85.1 \pm 0.2$ |
| VGAE | $93.3 \pm 0.6$ | $92.7 \pm 0.6$ | $80.1 \pm 0.0$ | $83.7 \pm 0.0$ |
| ARGA | $91.8 \pm 0.4$ | $90.7 \pm 0.4$ | $82.0 \pm 0.0$ | $85.5 \pm 0.1$ |
| ARGVA | $93.6 \pm 0.9$ | $93.0 \pm 1.0$ | $\underline{83.2 \pm 0.5}$ | $\underline{86.0 \pm 0.9}$ |
| GNAE | $94.1 \pm 0.2$ | $93.1 \pm 0.2$ | $82.8 \pm 2.1$ | $84.7 \pm 1.0$ |
| VGNAE | $93.2 \pm 0.3$ | $92.3 \pm 0.3$ | $77.3 \pm 0.1$ | $81.4 \pm 0.1$ |
| Bagging-PU | $96.0 \pm 0.1$ | $\underline{96.2 \pm 0.1}$ | $80.5 \pm 0.2$ | $85.0 \pm 0.1$ |
| PULL (ours) | $\mathbf{97.1 \pm 0.0}$ | $\mathbf{96.9 \pm 0.1}$ | $\mathbf{85.9 \pm 0.4}$ | $\mathbf{88.1 \pm 0.3}$ |

---

[4]https://github.com/shchur/gnn-benchmark/raw/master/data/npz/

