# OpenReview forum: "Accurate Link Prediction via PU Learning"
_ICLR.cc/2024/Conference — ICLR 2024 Conference Withdrawn Submission_

### Official Review · Reviewer_cg48 · 2023-10-26

**Soundness:** 3 good
**Presentation:** 3 good
**Contribution:** 2 fair
**Rating:** 6
**Confidence:** 4

**Summary:**

In this study, the authors tackle the issue of link prediction in edge-incomplete graphs. They introduce PULL, a novel method that applies positive-unlabeled (PU) learning, considering observed links as positive and non-observed ones as unlabeled. PULL iteratively labels
new edges with current link predictor. Extensive testing on real-world datasets confirms PULL's superiority over existing methods, marking a significant advancement in practical applications such as social media friend recommendations or academic citation analysis.

**Strengths:**

1. The authors study an important and practical problem as the fundamental assumption is that the un-present edges are not ALL negative.
2. The proposed method is simple and practical to implement on top of any link predictor.
3. The authors carry out empirical evaluation on five real-world dataset with comparison to several state-of-art methods.

**Weaknesses:**

1. Though the authors frame the problem and solution as PU learning, the proposed algorithm is an iterative method to carry out label augmentation. The technique have already been applied to several domains like CV and information retrieval.
2. The proposed hidden-variable framework is not necessary for the task if no edge potential on the link is assumed. It degenerates to a simple label augmentation with the current link predictor.
3. The empirical evaluation can be improved in following aspects: (1) As the proposed can be applied on top of any link predictor, it would be interesting to see whether the method can boost the performance of the baselines method; (2) Given the error bar, the proposed method is not statistical significant compared to baselines. Moreover, AUPRC would be a more sensitive metric under imbalanced labels. (3) All the graphs used in the experiments are small with ten of thousand of nodes. It would be interesting to see results on larger graphs.

**Questions:**

Please see weaknesses section.

---

> ### Author Response · Authors · 2023-11-19
> **Official Comment by Authors (1)**
>
> #### We sincerely thank you for your helpful feedback and insightful comments. We address your comments and questions below.
> #### In the revised draft, we mark our revisions in blue.
>
> > **[Q1] Though the authors frame the problem and solution as PU learning, the proposed algorithm is an iterative method to carry out label augmentation. The technique have already been applied to several domains like CV and information retrieval.**
> #### **[A1]** Thank you very much for this insightful suggestion. PULL stands apart from typical label augmentation strategies for the following three reasons.
> #### First, true negative examples (true unconnected node pairs) is not given in link prediction problem. We are given an edge-incomplete graph, which contains only observed links (a subset of ground-truth edges) while the other node pairs remain unlabeled. The absence of true unconnected node pairs poses a challenge in training the link predictor; it is not easy to train a binary classifier that classifies connected and unconnected node pairs given only the connected ones. Thus, previous label augmentation approaches, which assume a portion of true labels are given for every class, differ from the PU-learning-based approach of PULL.
> #### Second, in graph neural networks, there are two key components: 1) propagation scheme and 2) loss function. PULL changes both propagation scheme and loss function of previous methods. PULL propagates information through the expected graph structure with weights while the existing methods propagate information based on the given edge-incomplete graph. However, direct implementation of previous label augmentation approaches cannot change the propagation scheme since they aim to improve the quality of labels and incorporate them in the objective function, not in the propagation process.
> #### Finally, assigning predicted soft labels for unlabeled instances and incorporating them in loss function may seem similar to previous label augmentation methods. However, the loss $\mathcal{L}_E$ in PULL is not merely a simple loss using pseudo labels; instead, it is the expectation of negative log likelihoods for unconnected edges. This provides a robust theoretical foundation for PULL, establishing a connection with the EM (expectation-maximization) algorithm.
>
> > **[Q2] The proposed hidden-variable framework is not necessary for the task if no edge potential on the link is assumed. It degenerates to a simple label augmentation with the current link predictor.**
> #### **[A2]** We agree that PULL can be explained without those theoretical evidence. However, we believe that this is also the strength of PULL; it is intuitive but also has theoretical background. The proposed PULL is supported by theoretical foundations under the intuitive framework: 1) PULL propagates information through the *expectation of the graph structure*, where the quality is improved through the iterations, and 2) the iterative minimization of the proposed loss in PULL is similar to the iterative maximization of expected likelihood in the *EM (expectiation-maximization) algorithm*. The simplicity of PULL enhances the model's generalization capability, while those theoretical foundation ensures that PULL attains superior link prediction performance compared to the baselines.

---

> ### Author Response · Authors · 2023-11-19
> **Official Comment by Authors (2)**
>
> > **[Q3] The empirical evaluation can be improved in following aspects: (1) As the proposed can be applied on top of any link predictor, it would be interesting to see whether the method can boost the performance of the baselines method; (2) Given the error bar, the proposed method is not statistical significant compared to baselines. Moreover, AUPRC would be a more sensitive metric under imbalanced labels. (3) All the graphs used in the experiments are small with ten of thousand of nodes. It would be interesting to see results on larger graphs.**
> #### **[A3-1]** PULL can be applied to other GCN-based methods including GAE, VGAE, ARGA, and ARGVA that use GCN-based encoder. PULL cannot directly be applied to other baselines such as GAT, GraphSAGE, GNAE and VGNAE. This is because they use different propagation scheme instead of GCN, posing a challenge for PULL in propagating information through the expected graph structure during training. For example, GAT propagates information through the observed edges using the attention scores as weights. GraphSAGE performs random walk to define adjacent nodes. GNAE and VGNAE separate the feature mapping and propagation processes. We added an experiment that shows the performance improvements of those baselines when the framework of PULL is applied in Appendix F.1. Note that PULL improves the performance of the baselines in most of the cases, showing its applicability across various models.
> #### **[A3-2]** Thank you very much for your helpful suggestion. We added AUPRC score as our evaluation metric, and observed that PULL shows greater improvements in AUPRC compared to the baselines. We agree that in the original table, improvements seem to be marginal for some datasets. Nevertheless, we conducted the experiments ten times with random seeds, and the observed enhancements are significant in the field of graph mining, especially in terms of AUPRC score.
> #### **[A3-3]** We agree that the data used in our paper are not giant networks. In the field of link prediction, there have been limited analysis on giant networks until now; the baselines including GCN, GAT, GAE, VGAE, ARGA, ARVGA, GNAE, and VGNAE are evaluated on smaller graphs than ours in their works. GraphSAGE has used larger graphs than us, but GraphSAGE is a specially designed model for large graphs. Although previous approaches tend to evaluate their performance on small networks, we think the link prediction in larger graphs appears to be an intriguing subject. We added an experiment that performs link prediction in two larger graphs compared to those used in Section 4: ogbn-arxiv and Physics (see Appendix F.3). The ogbn-arxiv dataset is a citation network consisting of 169,343 nodes and 1,166,243 edges, where each node represents an arXiv paper and an edge indicates that one paper cites another one. Physics is a co-authorship graph based on the Microsoft Academic Graph from the KDD Cup 2016 challenge 3. Physics contains 34,493 nodes and 495,924 edges where each node represents an author, and they are connected if they co-authored a paper. Table 6 in Appendix F.3 shows that PULL consistently presents superior performance in terms of both AUROC and AUPRC. This indicates that PULL is also effective in handling larger graphs.
>
> >> **Table 6 in Appendix F.3 (when the missing ratio $r_m=0.1$).**
> | Model | Physics |  | ogbn-arxiv |  |
> |---|:---:|:---:|:---:|:---:|
> |  | AUROC | AUPRC | AUROC | AUPRC |
> | GCN+CE | 96.2 $\pm$ 0.0 | 95.9 $\pm$ 0.1 | 80.4 $\pm$ 0.3 | 84.7 $\pm$ 0.6 |
> | GAT+CE | 93.5 $\pm$ 0.1 | 92.1 $\pm$ 0.1 | 82.5 $\pm$ 0.2 | 79.9 $\pm$ 0.5 |
> | SAGE+CE | 95.8 $\pm$ 0.3 | 95.3 $\pm$ 0.3 | 82.4 $\pm$ 1.8 | 80.5 $\pm$ 1.2 |
> | GAE | 96.2 $\pm$ 0.1 | 95.8 $\pm$ 0.1 | 80.4 $\pm$ 0.3 | 85.1 $\pm$ 0.2 |
> | VGAE | 93.3 $\pm$ 0.6 | 92.7 $\pm$ 0.6 | 80.1 $\pm$ 0.0 | 83.7 $\pm$ 0.0 |
> | ARGA | 91.8 $\pm$ 0.4 | 90.7 $\pm$ 0.4 | 82.0 $\pm$ 0.0 | 85.5 $\pm$ 0.1 |
> | ARGVA | 93.6 $\pm$ 0.9 | 93.0 $\pm$ 1.0 | 83.2 $\pm$ 0.5 | 86.0 $\pm$ 0.9 |
> | GNAE | 94.1 $\pm$ 0.2 | 93.1 $\pm$ 0.2 | 82.8 $\pm$ 2.1 | 84.7 $\pm$ 1.0 |
> | VGNAE | 93.2 $\pm$ 0.3 | 92.3 $\pm$ 0.3 | 77.3 $\pm$ 0.1 | 81.4 $\pm$ 0.1 |
> | Bagging-PU | 96.0 $\pm$ 0.1 | 96.2 $\pm$ 0.1 | 80.5 $\pm$ 0.2 | 85.0 $\pm$ 0.1 |
> | PULL (ours) | **97.1 $\pm$ 0.0** | **96.9 $\pm$ 0.1** | **85.9 $\pm$ 0.4** | **88.1 $\pm$ 0.3** |

---

> > ### Comment · Reviewer_cg48 · 2023-11-23
> > **Thank you for your response**
> >
> > Thanks for authors' time and efforts on rebuttal. It addressed my concern especially on empirical evaluation part. I have updated my rating accordingly.

---

### Official Review · Reviewer_qtfB · 2023-10-31

**Soundness:** 3 good
**Presentation:** 3 good
**Contribution:** 3 good
**Rating:** 6
**Confidence:** 3

**Summary:**

The paper introduces a novel method called PULL, which leverages Positive-Unlabeled Learning (PU learning) to train link predictors more effectively. The approach is backed by both theoretical analysis and empirical tests. The article's experimentation encompasses five real-world datasets, and the results indicate that PULL significantly outperforms existing link prediction methods, achieving state-of-the-art performance. By addressing the inherent limitations of conventional techniques, the PULL method offers a more robust and accurate solution for link prediction in edge-incomplete graphs.

**Strengths:**

1．The PULL method employs latent variables to account for hidden relationships between unconnected node pairs, thereby effectively addressing the inherent uncertainty in edge-incomplete graphs. This is particularly advantageous as missing links are prevalent in real-world network environments.
2. The article demonstrates that the PULL method achieves state-of-the-art performance on five different real-world datasets, outperforming existing link prediction algorithms.
3. The inclusion of a theoretical analysis adds a layer of interpretability to the PULL method, enhancing its credibility and making it easier to understand its underlying mechanisms.

**Weaknesses:**

1. The PULL method incorporates latent variables and necessitates the construction of an expectation graph, potentially leading to high computational complexity. This could limit the method's scalability and applicability to large-scale datasets, requiring more computational power and time.

2. The selection of baselines for comparison could be broadened to include more recent methods, thereby providing a more comprehensive evaluation.

**Questions:**

Previous link prediction methods have relied too heavily on a given edge-incomplete graph and assumed that the edges of a given graph are all fully observed, without considering unobserved missing links. This approach causes the link predictor to overfit a given edge-incomplete graph, which reduces the accuracy of the prediction.

---

> ### Author Response · Authors · 2023-11-19
> **Official Comment by Authors**
>
> #### We thank the reviewer for the helpful comments and suggestions. We address your concerns as follows.
> #### We have also revised the paper accordingly and the revision is highlighted in blue color.
>
> > **[Q1] The PULL method incorporates latent variables and necessitates the construction of an expectation graph, potentially leading to high computational complexity. This could limit the method's scalability and applicability to large-scale datasets, requiring more computational power and time.**
> #### **[A1]** Thank you very much for the insightful suggestion. We agree that the construction of an expected graph $\bar{\mathcal{G}}$ potentially leads to high computational complexity since it has $O(|\mathcal{V}|^2 d)$ of time complexity where $d$ is the hidden feature size and $\mathcal{V}$ is the number of nodes in a graph. However, we think it is a minor weakness since such computation is common in many graph-based models such as HOG-GCN [1] and BM-GCN [2]. Furthermore, we approximated the expected graph to $\bar{\mathcal{G}}'$ by keeping the top-$K$ edges with the largest weights, while removing the rest. This ensures that the running time of PULL scales linearly with the number of edges (see Figure 3). We also added an experiment that evaluates PULL on larger networks compared to those used in Section 4 (see Appendix F.3).
>
> > **[Q2] The selection of baselines for comparison could be broadened to include more recent methods, thereby providing a more comprehensive evaluation.**
> #### **[A2]** Thank you very much for the valuable suggestion. We added a recent baseline, Bagging-PU [3], which incorporates PU learning for link prediction in graphs (see Table 1 of Section 4.2).
> >> **Comparison between PULL and Bagging-PU [1] (when the missing ratio $r_m=0.1$).**
> | Model | PubMed |  | Cora-full |  | Chameleon |  | Crocodile |  | Facebook |  |
> |---|:---:|:---:|:---:|:---:|:---:|:---:|:---:|:---:|:---:|:---:|
> |  | AUROC | AUPRC | AUROC | AUPRC | AUROC | AUPRC | AUROC | AUPRC | AUROC | AUPRC |
> | Bagging-PU | 94.6 $\pm$ 0.4 | 94.9 $\pm$ 0.4 | 92.6 $\pm$ 0.6 | 93.9 $\pm$ 0.5 | 97.4 $\pm$ 0.6 | 97.4 $\pm$ 0.7 | 97.4 $\pm$ 0.3 | 97.7 $\pm$ 0.3 | 97.0 $\pm$ 0.1 | 97.4 $\pm$ 0.1 |
> | PULL (ours) | **96.6 $\pm$ 0.2** | **96.9 $\pm$ 0.1** | **96.1 $\pm$ 0.3** | **96.4 $\pm$ 0.4** | **97.9 $\pm$ 0.2** | **97.9 $\pm$ 0.2** | **98.3 $\pm$ 0.1** | **98.5 $\pm$ 0.1** | **97.4 $\pm$ 0.1** | **97.7 $\pm$ 0.1** |
> #### Refer to Table 1 of Section 4.2 for full results.
>
> >  **[Q3] Previous link prediction methods have relied too heavily on a given edge-incomplete graph and assumed that the edges of a given graph are all fully observed, without considering unobserved missing links. This approach causes the link predictor to overfit a given edge-incomplete graph, which reduces the accuracy of the prediction.**
> #### **[A3]** It seems like the question is missing. Please clarify the question, and we will be happy to provide an answer.
>
> #### **[References]**
> #### [1] Wang, T., Jin, D., Wang, R., He, D., & Huang, Y. (2022). Powerful Graph Convolutional Networks with Adaptive Propagation Mechanism for Homophily and Heterophily. Proceedings of the AAAI Conference on Artificial Intelligence, 36(4), 4210-4218.
> #### [2] He, D., Liang, C., Liu, H., Wen, M., Jiao, P., & Feng, Z. (2022). Block Modeling-Guided Graph Convolutional Neural Networks. Proceedings of the AAAI Conference on Artificial Intelligence, 36(4), 4022-4029.
> #### [3] Shengfeng Gan, Mohammed Alshahrani, and Shichao Liu. Positive-unlabeled learning for network link prediction. Mathematics, 10(18):3345, 2022.

---

> > ### Comment · Reviewer_qtfB · 2023-11-23
> >
> > Dear authors,
> > Thank you for your response and pardon my late reply. I appreciate the time and efforts you put on rebuttal. Most of my concerns have been addressed in your response.

---

### Official Review · Reviewer_SEg5 · 2023-11-01

**Soundness:** 4 excellent
**Presentation:** 4 excellent
**Contribution:** 3 good
**Rating:** 8
**Confidence:** 3

**Summary:**

This paper proposes PULL (PU-LEARNING-BASED LINK PREDICTOR), an accurate method for link prediction in edge-incomplete graphs. The PULL proposed with the thought that in real-world scenarios the presence of missing edges is frequently observed. Under this assumption, without the consideration of edge-incomplete graph, it may degrade the link prediction performance. Thus, it is important to consider the uncertainties of the given graph to obtain accurate linking probabilities between nodes. Expecting to propose PULL, this paper conducts a theoretical analysis of PULL, studying its relationship with the EM algorithm. The PULL achieves state-of-the-art link prediction performance. In summary, the contributions of this paper are as follows:
	Proposing PULL, an accurate method for link prediction in graphs
	Conduct a theoretical analysis of PULL, studying its relationship with the EM algorithm.
	Through the experiments on five datasets, PULL achieves state-of-the-art link prediction performance
The manuscript demonstrates notable originality with the introduction of a new method named PULL. This method has potential implications in the domain of graph data mining, which could be of considerable significance. In terms of linguistic presentation, the article is well-composed. The flow between sentences appears logical, making the content coherent and accessible to the reader. The authors have taken care to provide a structured proof of PULL, which strengthens the paper's theoretical grounding. Regarding the experimental section, the authors have provided a detailed description of their experiments, followed by an in-depth analysis of the data.
Overall, this paper could be a significant algorithmic contribution, I would be willing to increase the score.

**Strengths:**

Introduce the consideration of edge-incomplete graph, which has a significant contribution
The theoretical analysis and extensive experiments support the claims of the paper
The authors provide the code and datasets, enhancing the reproducibility of the results

**Weaknesses:**

The paper could provide more detailed explanations of the experimental setup, such as the hyperparameter settings and the choice of evaluation metrics.

**Questions:**

How does the proposed method handle large-scale graphs？ Are there any scalability issues？
Are there any limitations or potential drawbacks of the proposed method that need to be considered？
Can the proposed method be extended to handle multi-relational graphs？

---

> ### Author Response · Authors · 2023-11-19
> **Official Comment by Authors**
>
> #### We thank the reviewer for their positive feedback. We address your questions as follows.
> #### We have also revised the paper accordingly and the revision is highlighted in blue color.
>
> > **[Q1] The paper could provide more detailed explanations of the experimental setup, such as the hyperparameter settings and the choice of evaluation metrics.**
> #### **[A1]** Thank you very much for your suggestion. We added detailed hyperparameter settings in Appendix E. For the evaluation metric, we use AUROC (area under the ROC curve) score following many other studies for link prediction in graphs [1,2]. AUC score is widely used in link prediction problem for two reasons: 1) it is robust to class imbalance, and 2) it assesses how well a model can rank instances by their predicted scores, where ranking edges is crucial in predicting links. We also added AUPRC (area under the precision-recall curve) score as a metric for a thorough evaluation. We conducted experiments ten times using ten different random seeds for each dataset, and we report the average score along with the standard deviation.
>
> > **[Q2] How does the proposed method handle large-scale graphs? Are there any scalability issues?**
> #### **[A2]** PULL handles large-scale graphs by utilizing an approximated one of $\bar{\mathcal{G}}$ for training the link predictor. Specifically, PULL approximates $\bar{\mathcal{G}}$ by keeping the top-$K$ edges with the largest weights, while removing the rest. This ensures that the running time of PULL scales linearly with the number of edges (see Figure 3). We also added an experiment that evaluates PULL in larger networks compared to those used in Section 4 (see Appendix F.3).
>
> > **[Q3] Are there any limitations or potential drawbacks of the proposed method that need to be considered?**
> #### **[A3]** Changing the labels and structures of PULL may require the link predictor to adapt to the new labels and the structure, potentially leading to additional time for training. However, the proposed PULL aims to train an accurate link predictor, and the experimental results in Table 1 show that PULL consistently presents superior accuracy than the previous approaches. Despite the potential drawback of PULL, we believe that the concept of considering missing edges during training has the potential to advance existing approaches, eliminating the assumption that given edges are fully observed, thereby expanding the coverage of applications towards more realistic scenarios. We added the potential drawback in the conclusion section (Section 5).
>
> > **[Q4] Can the proposed method be extended to handle multi-relational graphs?**
> #### **[A4]** Thank you for proposing exciting avenues for future research. It is not easy for PULL to be directly applied to multi-relational graphs since PULL assumes homophily between nodes, predicting links based on similarities in hidden node representations. In multi-relational graphs, like signed graphs, nodes with distinct representations also form negative edges. Nevertheless, exploring the extension of PULL to handle such complexities would be an intriguing avenue for further investigation. We added the discussion in the conclusion section (Section 5).
>
> #### **[References]**
> #### [1] Shirui Pan, Ruiqi Hu, Guodong Long, Jing Jiang, Lina Yao, and Chengqi Zhang. Adversarially regularized graph autoencoder for graph embedding. In IJCAI, pp. 2609–2615. ijcai.org, 2018.
> #### [2] Seong-Jin Ahn and Myoung Ho Kim. Variation al graph normalized autoencoders. In CIKM, pp. 2827–2831. ACM, 2021.

---

### Official Review · Reviewer_LBxh · 2023-11-04

**Soundness:** 3 good
**Presentation:** 4 excellent
**Contribution:** 3 good
**Rating:** 8
**Confidence:** 4

**Summary:**

The authors propose the PULL method for link prediction using positive-unlabeled (PU) learning. The basic idea is to treat all edges as positive samples while treating all non-edges as unlabeled rather than negative samples. They then use a graph convolutional network (GCN) as a link predictor to predict edge probabilities for all the non-edges (unlabeled samples) and turn the ones with the highest probabilities into edges for the next iteration of learning. (In this sense, it shares some similarities with self learning, where predictions are used as labels to re-train a classifier.) The authors further demonstrate an equivalence between their learning algorithm and the well-known expectation-maximization (EM) algorithm under an independence assumption. They show improved link prediction accuracy on several real data sets and demonstrate that they typically achieve maximum accuracy at around the true number of edges in a graph.

*After author rebuttal:* The authors have done a thorough job of answering the reviewers' questions and addressing their concerns with a much improved revision. I have raised my score to 8 to reflect the improvement in quality.

**Strengths:**

- Creative approach using PU learning for link prediction, a task that seems to be an ideal fit for the assumptions of PU learning.
- Detailed empirical investigation of several research questions that go beyond improvement in accuracy. I found the analysis of accuracy both with and without the $\mathcal{L}_C$ term as a function of the number of iterations to be very insightful.
- Well written paper that was easy to understand and explains the PULL method with an appropriate level of detail.

**Weaknesses:**

- While the overall framework is quite principled, some ad-hoc tweaks seem to be necessary to get it working well, such as the way the number of selected edges $K$ is increased or the need for the additional loss function $\mathcal{L}_C$.
- ~~There is some other related work on PU learning for link prediction that is not cited (Gan et al., 2022; chapter 4 of Hao, 2021). While they also use PU learning, the approaches seem to be different from what the authors propose, so I don't think it limits the novelty of this paper and find it to be only a minor weakness.~~ This weakness was addressed by the authors in their revision during the discussion phase and no longer applies.

References:
- Gan, S., Alshahrani, M., & Liu, S. (2022). Positive-Unlabeled Learning for Network Link Prediction. Mathematics, 10(18), 3345.
- Hao, Y. (2021). Learning node embedding from graph structure and node attributes (Doctoral dissertation, UNSW Sydney).

**Questions:**

1. Hao (2021) present their PU learning-based link predictor as a wrapper that could be applied to a variety of different GNNs. Could your proposed PULL approach generalize beyond the GCN that you use in this paper? If so, I think that would be a better comparison--apply your PU learning approach to each GNN and then compare the accuracy of your PU-based model with the normally trained model.
2. Why keep the top $K$ with the highest predicted probabilities? It seems like taking some sort of random sample based on the probabilities might be a better approach to avoid too much self reinforcement, which could lead to oversmoothing. Taking a random sample could eliminate the need for the $\mathcal{L}_C$ term.

---

> ### Author Response · Authors · 2023-11-19
> **Official Comment by Authors (1)**
>
> #### We thank the reviewer for the insightful feedback. We address your questions as follows.
> #### We have also revised the paper accordingly and the revision is highlighted in blue color.
>
> > **[Q1] While the overall framework is quite principled, some ad-hoc tweaks seem to be necessary to get it working well, such as the way the number of selected edges $K$ is increased or the need for the additional loss function $\mathcal{L}_C$.**
>
> #### **[A1]**  We agree that the way of increasing the number $K$ of selected edges, and introducing the additional loss function $\mathcal{L}_C$ may appear somewhat arbitrary. However, we argue that these tweaks are grounded in intuition.
> #### First, PULL increases the number $K$ of selected edges through the iterations, reflecting an inherent belief that the quality of the expected graph structure improves as the iterations progress. The approach of increasing the number $K$ gives more trust in the expected graph structure $\bar{\mathcal{G}}$ by selecting more edges from the pool of unconnected node pairs $\mathcal{E}_\mathcal{U}$. Furthermore, increasing the number $K$ in proportion to that of observed edges is effective in approximating the real number of edges in the ground-truth graph, as illustrated in Figure 2. We added explanations for increasing the number $K$ in Section 3.2.
> #### Second, the incorporation of the additional loss term $\mathcal{L}_C$ effectively prevents too much self-reinforcement tendencies within the link predictor of PULL. Especially, $\mathcal{L}_C$ enables PULL to preserve its accuracy when there are more number of edges in the approximated $\bar{\mathcal{G}}'$ of the expected graph $\bar{\mathcal{G}}$. If the current link predictor is trained on $\bar{\mathcal{G}}'$ consisting of excessive number of edges without $\mathcal{L}_C$, the quality of newly computed $\bar{\mathcal{G}}$ deteriorates, leading to the next iteration's link predictor becoming even more inaccurate. This case is illustrated in Figure 4. We have provided additional clarification for the additional loss term in Section 3.3 of the revised paper.
>
> > **[Q2] There is some other related work on PU learning for link prediction that is not cited (Gan et al., 2022; chapter 4 of Hao, 2021). While they also use PU learning, the approaches seem to be different from what the authors propose, so I don't think it limits the novelty of this paper and find it to be only a minor weakness.**
> #### **[A2]** Thank you very much for making us aware of this important prior work. We added those references in Section 2.2. We agree that they are also PU-learning-based approaches for link prediction in graphs. However, their performance is limited since they propagate information through the given edge-incomplete graph for obtaining node and edge representations. We also added the work of Gan et al. (2022) as our baseline in experiments (see Table 1 of Section 4.2).

---

> > ### Comment · Reviewer_LBxh · 2023-11-23
> > **Thank you for the detailed revision**
> >
> > I have increased my score to 8 based on the quality of your revision and hope to see this paper accepted!

---

> ### Author Response · Authors · 2023-11-19
> **Official Comment by Authors (2)**
>
> > **[Q3] Hao (2021) present their PU learning-based link predictor as a wrapper that could be applied to a variety of different GNNs. Could your proposed PULL approach generalize beyond the GCN that you use in this paper? If so, I think that would be a better comparison--apply your PU learning approach to each GNN and then compare the accuracy of your PU-based model with the normally trained model.**
> #### **[A3]** We express sincere gratitude for your suggestion. PULL can be applied to other GCN-based methods including GAE, VGAE, ARGA, and ARGVA that use GCN-based encoder. PULL cannot directly be applied to other baselines such as GAT, GraphSAGE, GNAE and VGNAE. This is because they use different propagation scheme instead of GCN, posing a challenge for PULL in propagating information through the expected graph structure during training. For example, GAT propagates information through the observed edges with attention scores as weights. GraphSAGE performs random walk to define adjacent nodes. GNAE and VGNAE separate the feature mapping and propagation processes. We added an experiment that shows the performance improvements of the baselines when the framework of PULL is applied (see Appendix F.1). Note that PULL improves the performance of the baselines in most of the cases, showing its applicability across various models.
>
> >> **Table 4 in Appendix F.1 (when the missing ratio $r_m=0.1$)**
> | Model | PubMed |  | Cora-full |  | Chameleon |  | Crocodile |  | Facebook |  |
> |---|:---:|:---:|:---:|:---:|:---:|:---:|:---:|:---:|:---:|:---:|
> |  | AUROC | AUPRC | AUROC | AUPRC | AUROC | AUPRC | AUROC | AUPRC | AUROC | AUPRC |
> | GAE | 96.0 $\pm$ 0.1 | 95.9 $\pm$ 0.2 | 95.1 $\pm$ 0.5 | 95.1 $\pm$ 0.4 | 96.4 $\pm$ 0.2 | 96.3 $\pm$ 0.2 | 95.8 $\pm$ 0.6 | 96.2 $\pm$ 0.6 | 96.6 $\pm$ 0.2 | 96.9 $\pm$ 0.1 |
> | GAE+PULL | **96.5 $\pm$ 0.1** | **96.7 $\pm$ 0.2** | **95.4 $\pm$ 0.5** | **95.6 $\pm$ 0.5** | **97.8 $\pm$ 0.2** | **97.8 $\pm$ 0.2** | **97.5 $\pm$ 0.4** | **97.3 $\pm$ 0.4** | **97.2 $\pm$ 0.3** | **97.5 $\pm$ 0.2** |
> | VGAE | 93.8 $\pm$ 1.3 | 93.7 $\pm$ 1.1 | 91.5 $\pm$ 3.0 | 88.1 $\pm$ 5.3 | 95.9 $\pm$ 0.6 | 95.9 $\pm$ 0.2 | 94.7 $\pm$ 0.3 | 94.3 $\pm$ 0.9 | 94.3 $\pm$ 0.9 | 94.5 $\pm$ 0.4 |
> | VGAE+PULL | **95.2 $\pm$ 0.4** | **95.2 $\pm$ 0.3** | **93.8 $\pm$ 0.3** | **93.9 $\pm$ 0.4** | **97.0 $\pm$ 0.0** | **97.1 $\pm$ 0.1** | **96.3 $\pm$ 0.2** | **96.3 $\pm$ 0.2** | **96.4 $\pm$ 0.3** | **96.6 $\pm$ 0.3** |
> | ARGA | 93.2 $\pm$ 0.7 | 93.1 $\pm$ 0.4 | 90.4 $\pm$ 1.0 | 90.2 $\pm$ 0.5 | 94.8 $\pm$ 0.4 | 94.7 $\pm$ 0.3 | 95.9 $\pm$ 0.5 | 95.5 $\pm$ 0.6 | 91.8 $\pm$ 0.7 | 91.8 $\pm$ 0.2 |
> | ARGA+PULL | **93.9 $\pm$ 0.6** | **94.6 $\pm$ 0.5** | **94.4 $\pm$ 1.0** | **93.9 $\pm$ 1.2** | **96.3 $\pm$ 0.3** | **96.0 $\pm$ 0.3** | **95.2 $\pm$ 0.6** | **95.7 $\pm$ 0.7** | **93.5 $\pm$ 0.6** | **93.8 $\pm$ 0.6** |
> | ARGVA | 93.5 $\pm$ 1.2 | 93.4 $\pm$ 0.4 | 88.2 $\pm$ 3.8 | 84.3 $\pm$ 4.3 | 93.6 $\pm$ 0.5 | 93.8 $\pm$ 0.2 | 94.9 $\pm$ 0.3 | 94.1 $\pm$ 0.1 | 92.5 $\pm$ 2.4 | 92.7 $\pm$ 1.8 |
> | ARGVA+PULL | **94.8 $\pm$ 0.3** | **94.9 $\pm$ 0.4** | **93.7 $\pm$ 0.4** | **93.6 $\pm$ 0.5** | **95.3 $\pm$ 0.3** | **95.1 $\pm$ 0.0** | **95.4 $\pm$ 0.3** | **95.5 $\pm$ 0.3** | **94.8 $\pm$ 0.2** | **95.2 $\pm$ 0.1** |

---

> ### Author Response · Authors · 2023-11-19
> **Official Comment by Authors (3)**
>
> > **[Q4] Why keep the top $K$ with the highest predicted probabilities? It seems like taking some sort of random sample based on the probabilities might be a better approach to avoid too much self reinforcement, which could lead to oversmoothing. Taking a random sample could eliminate the need for the $\mathcal{L_C}$ term.**
> #### **[A4]** Thank you very much for this helpful suggestion. We agree that the $\mathcal{L_C}$ term prevents the link predictor from too much self reinforcement, and the random-sampling-based approach might eliminate the need for $\mathcal{L_C}$. We added an experiment that evaluates PULL with random sampling of $K$ links from the expected graph $\bar{\mathcal{G}}$ based on the linking probabilities, instead of the top-$K$ links (see Appendix F.2). As the weighted random sampling empowers PULL to mitigate the excessive self-reinforcement in the link predictor, we exclude the loss term $\mathcal{L}_C$, which serves the same purpose. Table 5 in Appendix F.2 shows that the proposed PULL presents higher prediction performance than the sampling-based PULL-WS (PULL with Weighted Sampling) in most of the cases. However, PULL-WS is an efficient variant of PULL that uses only a single loss term $\mathcal{L}'_E$ instead of the proposed loss $\mathcal{L} = \mathcal{L}_E' + \mathcal{L}_C$. Thank you again for sharing the valuable idea.
>
> >> **Table 5 in Appendix F.2 (when the missing ratio $r_m=0.1$)**
> | Model | PubMed |  | Cora-full |  | Chameleon |  | Crocodile |  | Facebook |  |
> |---|:---:|:---:|:---:|:---:|:---:|:---:|:---:|:---:|:---:|:---:|
> |  | AUROC | AUPRC | AUROC | AUPRC | AUROC | AUPRC | AUROC | AUPRC | AUROC | AUPRC |
> | PULL-WR | 96.5 $\pm$ 0.2 | 96.7 $\pm$ 0.1 | 96.0 $\pm$ 0.4 | 96.3 $\pm$ 0.4 | 97.4 $\pm$ 0.1 | 97.5 $\pm$ 0.2 | 97.6 $\pm$ 0.1 | 98.0 $\pm$ 0.1 | 97.2 $\pm$ 0.1 | 97.6 $\pm$ 0.1 |
> | PULL (ours) | **96.6 $\pm$ 0.2** | **96.9 $\pm$ 0.1** | **96.1 $\pm$ 0.3** | **96.4 $\pm$ 0.4** | **97.9 $\pm$ 0.2** | **97.9 $\pm$ 0.2** | **98.3 $\pm$ 0.1** | **98.5 $\pm$ 0.1** | **97.4 $\pm$ 0.1** | **97.7 $\pm$ 0.1** |